# A simeprevir-inducible molecular switch for the control of cell and gene therapies

Stacey E. Chin [1], Christina Schindler [1], Lisa Vinall[1], Roger B. Dodd [2], Lisa Bamber[2], Sandrine Legg[2], Anna Sigurdardottir [2], D. Gareth Rees [2], Tim I. M. Malcolm[1], Samantha J. Spratley[2], Cecilia Granéli [3], Jonathan Sumner[1] & Natalie J. Tigue [1] ✉

Chemical inducer of dimerization (CID) modules can be used effectively as molecular switches to control biological processes, and thus there is significant interest within the synthetic biology community in identifying novel CID systems. To date, CID modules have been used primarily in engineering cells for in vitro applications. To broaden their utility to the clinical setting, including the potential to control cell and gene therapies, the identification of novel CID modules should consider factors such as the safety and pharmacokinetic profile of the small molecule inducer, and the orthogonality and immunogenicity of the protein components. Here we describe a CID module based on the orally available, approved, small molecule simeprevir and its target, the NS3/4A protease from hepatitis C virus. We demonstrate the utility of this CID module as a molecular switch to control biological processes such as gene expression and apoptosis in vitro, and show that the CID system can be used to rapidly induce apoptosis in tumor cells in a xenograft mouse model, leading to complete tumor regression.

The identification of component parts to enable the engineering of cellular pathways and processes is a primary goal of synthetic biology. Chemical inducers of dimerization systems have been used extensively for reversible and tuneable regulation of processes dependent on protein proximity. Early work focused on the natural compound rapamycin which induces dimerization of FKBP and FBP, and although this system is now widely used in cell biology applications, rapamycin is not bioorthogonal in mammalian cells where its natural target is present[1]. Furthermore, the toxicity and immunosuppressive properties of rapamycin can complicate its use as a CID[2]. Less toxic rapamycin analogues are available but their use is limited by poor PK[3]. Other natural CIDs have also been exploited for synthetic biology applications, including gibberellin[4] and abscisic acid[5], which induce dimerization of their natural target proteins; these CID modules are plant-derived and thus are bioorthogonal in mammalian cells. These compounds, however, have not been tested in the clinic,

which presents a significant regulatory hurdle for use in therapeutic applications.

More recently, several groups have identified novel CID modules based on synthetic or natural compounds, using protein engineering, phage display, and/or computational design to identify the protein components of the CID system[6]. In one example, computational design has been used to introduce new ligand-binding pockets at existing protein–protein interfaces[7]. In another study, two sequential phage display selections were used to identify both protein components of a CID system[8]. An alternative approach is to start from a known protein:small molecule complex and identify specific protein binding partners by selection from phage display libraries of antibody (or antibody-like scaffold) proteins[9–11] or computational design[12]. Protein engineering and directed evolution have also been used to alter the ligand-binding specificity of the natural ABA-based CID, thus creating a variety of CID modules[13–15]. These systems have been used primarily

[1]Discovery Sciences, BioPharmaceuticals R&D, AstraZeneca, Cambridge, UK. [2]Biologics Engineering, Oncology R&D, AstraZeneca, Cambridge, UK. [3]Bio-Pharmaceuticals R&D Cell Therapy Department, Research and Early Development, Cardiovascular, Renal, and Metabolism (CVRM), BioPharmaceuticals R&D, AstraZeneca, Gothenburg, Sweden. ✉e-mail: natalie.tigue@astrazeneca.com

in vitro, as biosensors, or to regulate protein proximity-dependent cellular processes such as cellular co-localization, chimeric antigen receptor-T cell (CAR T) activation, and expression of endogenous and exogenous genes.

Here, we set out to identify a CID module with properties to enable rapid translation from in vitro applications to potential use in humans. In particular, we focused on small molecules that are clinically approved, have no endogenous human target, and have favorable PK and safety profiles. We describe a CID system based on the clinically approved antiviral simeprevir and its target, the NS3/4A protease (PR) from the hepatitis C virus (HCV). We demonstrate that this CID module can be used to regulate expression of heterologous genes delivered by transient transfection and via adeno-associated virus (AAV), to control expression of an endogenous gene, and to induce apoptosis in vitro. Importantly, we demonstrate that a single dose of simeprevir induces complete regression in a tumor-bearing mouse model when the CID module is incorporated into a kill switch.

## Results

### Identification of a chemical inducer of dimerization module based on simeprevir and HCV NS3/4A PR

To generate a de novo chemical inducer of dimerization (CID) module consisting of two proteins and a small molecule inducer (Fig. 1a), we first identified suitable inducer/protein 1 combinations based on existing clinically approved small molecules and their targets. Protein 2 candidates that selectively bind to the complex could then be selected from an antibody-like scaffold library using phage display technology. We prioritized antiviral compounds and their targets over approved small molecules with human targets to avoid on-target pharmacology and issues associated with an endogenous target sink. In addition, unlike compounds targeting other non-human proteins (e.g. antibiotics), antivirals are typically better tolerated in chronic dosing settings. Furthermore, we chose to avoid overexpression of any human target proteins in a cellular context which could have unexpected consequences due to intrinsic activity and interaction with their natural ligands.

Taking these factors into consideration, we chose the small molecule/protein pair of simeprevir and its target, the NS3/4A protease from hepatitis C virus (HCV NS3/4A PR) as the basis for a CID module. Simeprevir (Olysio®) is an approved small molecule that is administered orally, is cell-permeable, and has a pharmacokinetic (PK) profile that supports once-daily dosing in humans[16]. The HCV non-structural protein 3 (NS3) comprises both a protease and a helicase, but simeprevir binds to the protease domain alone. The protease domain can be expressed in isolation as a small (21 kDa), monomeric protein stabilized by solubilizing mutations and fusion to a short peptide from the cofactor protein NS4A. This construct is suitable for cytoplasmic expression and is not found associated with DNA, unlike many antiviral targets. Furthermore, three-dimensional X-ray crystallography of the complex (PDB code: 3KEE) reveals that simeprevir is bound in the shallow substrate-binding groove of HCV NS3/4A PR with 364 Å of exposed surface area (Fig. 1b)[17]. We reasoned that this relatively large exposed area would be sufficiently different from the unbound HCV NS3/4A PR such that complex-specific binding molecules could be identified.

The HCV NS3/4A PR is an enzyme that cleaves at four junctions of the HCV polyprotein precursor, and it is known to cleave a limited number of endogenous human targets[18,19]. To limit this activity within human cells, we used a single residue active site variant of HCV NS3/4A PR (S139A) that was previously shown to demonstrate significantly less activity than its wild-type counterpart[20] (Supplementary Fig. 1a). We used a fluorogenic peptide cleavage assay to confirm that this variant is enzymatically inactive (Supplementary Fig. 1b) and isothermal calorimetry to confirm that this variant retains high affinity binding to simeprevir (Supplementary Fig. 1c). Based on these data we chose to

proceed with the selection of HCV NS3/4A PR:simeprevir complex-specific binding (PRSIM) molecules against the S139A mutant protein. For the sake of simplicity, hereafter we shall refer to the S139A mutant of HCV NS3/4A PR as HCV PR.

PRSIM molecules were isolated by phage display selections from a Tn3 library developed as an alternative antibody-like scaffold based on the third FnIII module in human tenascin C[21–23]. Four rounds of phage display selections were performed on biotinylated HCV PR in the presence of a 50-fold molar excess of simeprevir. Phage ELISA was used to identify individual Tn3 molecules that bound to biotinylated HCV PR in the presence of simeprevir, but not to biotinylated HCV PR alone (Supplementary Fig. 2). A panel of 28 Tn3s with unique sequences in the three variable loops that demonstrated selective binding to biotinylated HCV PR in the presence of simeprevir in the phage ELISA (Supplementary Table 1) were selected for further biochemical studies using a homogeneous time-resolved fluorescence (HTRF) binding assay to measure ternary complex formation. A panel of 8 Tn3s was confirmed as complex-specific with no detectable binding to HCV PR in the absence of simeprevir (Fig. 1c; Supplementary Fig. 3). We also used this HTRF assay to investigate whether simeprevir binds to any significant extent to PRSIM_23 alone. HCV PR and PRSIM_23 were fixed at concentrations of 5 nM and 6 nM, respectively, and simeprevir was titrated up to a maximum concentration of 7.4 μM (Supplementary Fig. 4). At simeprevir concentrations greater than the concentration of HCV PR in this assay, HCV PR will be saturated by simeprevir, and an excess of unbound simeprevir will be available in solution. If simeprevir interacts with PRSIM_23 to any significant extent, then free simeprevir should compete for binding to PRSIM_23 with the HCV PR:simeprevir complex, and a concomitant decrease in complex formation should be observed. However, at concentrations up to three orders of magnitude greater than the EC50 in this assay, we observed no decrease in ternary complex formation, suggesting that PRSIM_23 does not interact with simeprevir when it is not bound to HCV PR.

The kinetics of binding of defined concentrations of HCV PR to six Tn3s in the presence or absence of simeprevir were determined using surface plasmon resonance (SPR). Five of these Tn3s showed strong and selective binding to simeprevir-bound HCV PR and only one (PRSIM_48) showed significant non-specific binding to HCV PR alone (Supplementary Table 2; Supplementary Fig. 5). PRSIM_23 showed the highest level of binding (24% of theoretical $R_{max}$) in the presence of 10 nM simeprevir and the HCV PR:simeprevir complex bound to PRSIM_23 with an apparent affinity constant ($K_{D, app}$) of 6.1 nM (Fig. 1d; Supplementary Table 2). By contrast, in the absence of simeprevir, PRSIM_23 displayed only weak binding (6.2 μM) to HCV PR as measured by SPR (Fig. 1e; Supplementary Table 3). A key parameter for describing the activity of CID-like compounds is cooperativity (α), which in this system corresponds to the ratio of the affinities of PRSIM_23 for HCV_PR in the absence and presence of simeprevir; based on the results shown here, the cooperativity conferred by simeprevir to the HCV PR:PRSIM_23 interaction is at least 1000-fold, which compares favorably to other described CID modules[24]. Thus, simeprevir can induce significant levels of ternary complex formation at protein concentrations where no binding between HCV PR and PRSIM_23 is observed in the absence of simeprevir (Supplementary Fig. 6). The effect of simeprevir concentration on the formation of the HCV PR:PRSIM_23 complex was also measured using SPR; simeprevir had an EC50 for PRSIM_23 in complex of 4.0 nM (Fig. 1f).

Having demonstrated that the formation of the active switch complex between HCV PR and PRSIM_23 is dependent on the presence of simeprevir, we wanted to test the specificity of this interaction with respect to alternative small molecule inhibitors of HCV PR. A panel of clinically approved HCV PR inhibitors (asunaprevir, boceprevir, danoprevir, glecaprevir, grazoprevir, narlaprevir, paritaprevir, telaprevir, and vaniprevir[25]) was assessed for their ability to induce

complex formation between HCV PR and PRSIM_23 in an HTRF binding assay. We found that induction of ternary complex formation was specific for simeprevir as none of the HCV PR inhibitors (Fig. 1g, Supplementary Fig. 7) could form a complex with HCV PR and PRSIM_23.

**PRSIM-based CID modules can regulate gene expression via reconstitution of a split transcription factor**

Having isolated PRSIM molecules that specifically bound HCV PR:simeprevir complexes, we reasoned that the system could be used to regulate the expression of transgenes via the fusion of the two protein

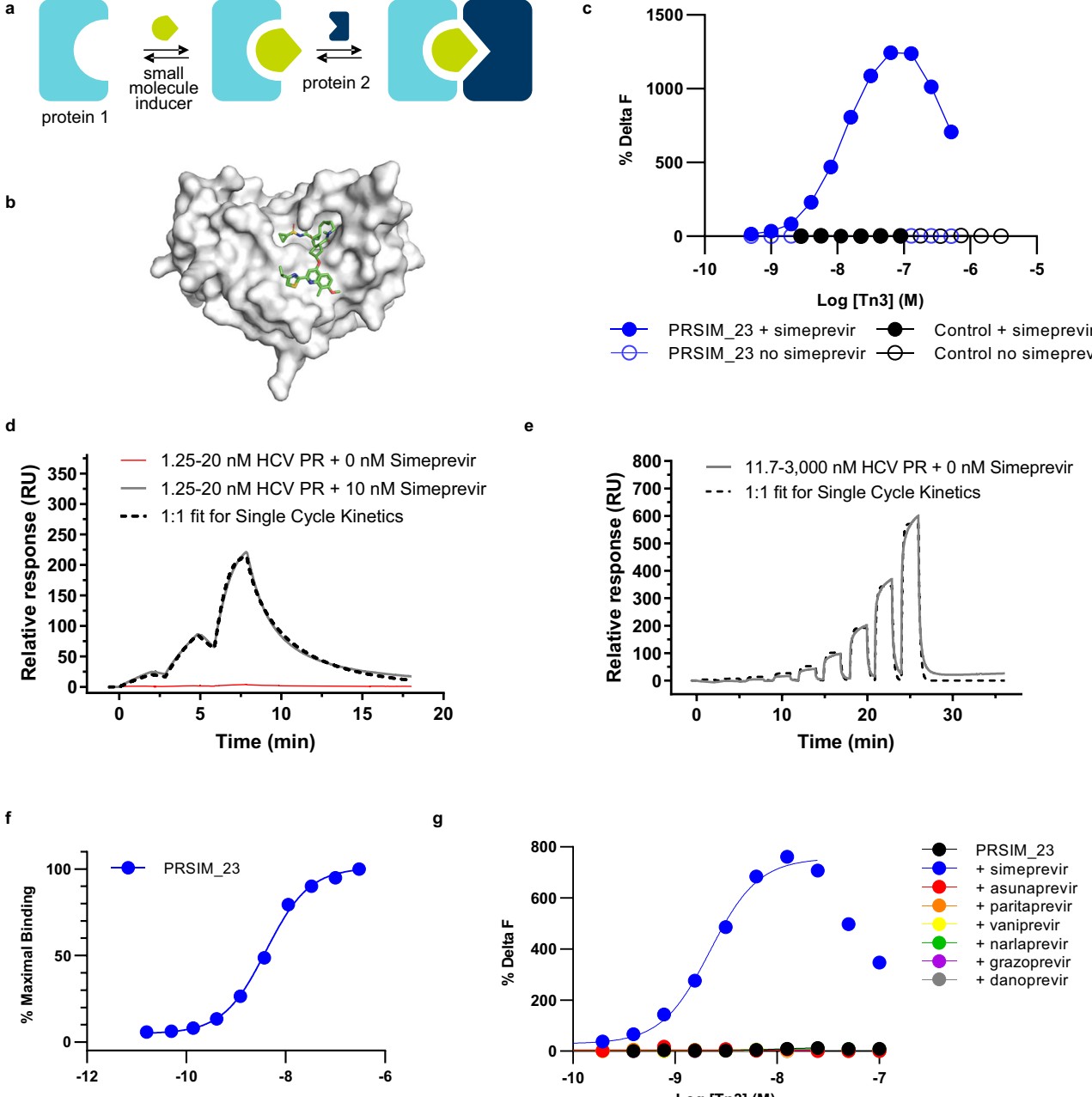

**Fig. 1 | Identification and characterization of HCV NS3/4A PR:simeprevir (PRSIM)-based chemical inducer of dimerization (CID) modules. a** Schematic of PRSIM modules. **b** The crystal structure of the HCV PR:simeprevir complex (PBD code: 3KEE) shows that simeprevir binds in a shallow surface groove of HCV PR with a large surface area of simeprevir remaining solvent exposed. The structure is visualized using The PyMOL Molecular Graphics System Version 2.1.0 (Schrödinger). **c** PRSIM_23 binds to HCV PR in complex with simeprevir (blue closed circles) but not to HCV alone (blue open circles) in a homogenous time-resolved fluorescence (HTRF) assay. The observed assay signal decrease at high PRSIM_23 concentrations is likely due to depletion of detection reagent. No binding of a control Tn3 (black circles) to HCV PR is observed. Each data point represents the mean of two independent replicates. **d** HCV PR binds PRSIM_23 in the presence of 10 nM simeprevir with an apparent affinity of 6.1 nM, in a Surface Plasmon Resonance

(SPR) assay. No binding of HCV PR alone is seen at the same HCV PR concentration. **e** HCV PR shows low-affinity binding (6.2 μM) to PRSIM_23 in the absence of simeprevir. **f** Dose-dependent heterodimerization of HCV PR and PRSIM_23 was induced by simeprevir in an SPR assay with an EC50 of 4.0 nM. Each data point represents the mean ± s.d. of three independent replicates. The dose−response curve was fit to the data using 3 parameter nonlinear regression. **g** PRSIM_23 binds to HCV PR in complex with simeprevir (blue) but not to HCV PR in complex with other HCV protease inhibitors in an HTRF assay. Each data point represents the mean of two independent replicates. The dose−response curve was fit to the data using 4 parameter nonlinear regression excluding the data points at the two highest concentrations of simeprevir where detection reagent depletion is observed. Source data for **c**−**g** are provided as a Source Data file.

components to the two domains of a split transcription factor. To demonstrate this, we used the iDimerize regulated transcription system (Takara) that is based on the rapamycin-inducible FK506-binding protein (FKBP):FKBP−rapamycin binding domain (FRB) interaction which brings together the activation domain (AD; p65) and DNA-binding domain (DBD; ZFHD1) of a split transcription factor. When the

rapalog inducer (AP21967) is added to cells transiently expressing the two components, the full-length transcription factor is functionally reconstituted and transgene expression from an inducible promoter is initiated.

We exchanged the FRB and FKBP12 coding sequences in the system for those encoding the HCV PR and one copy of PRSIM_23 (Fig. 2a)

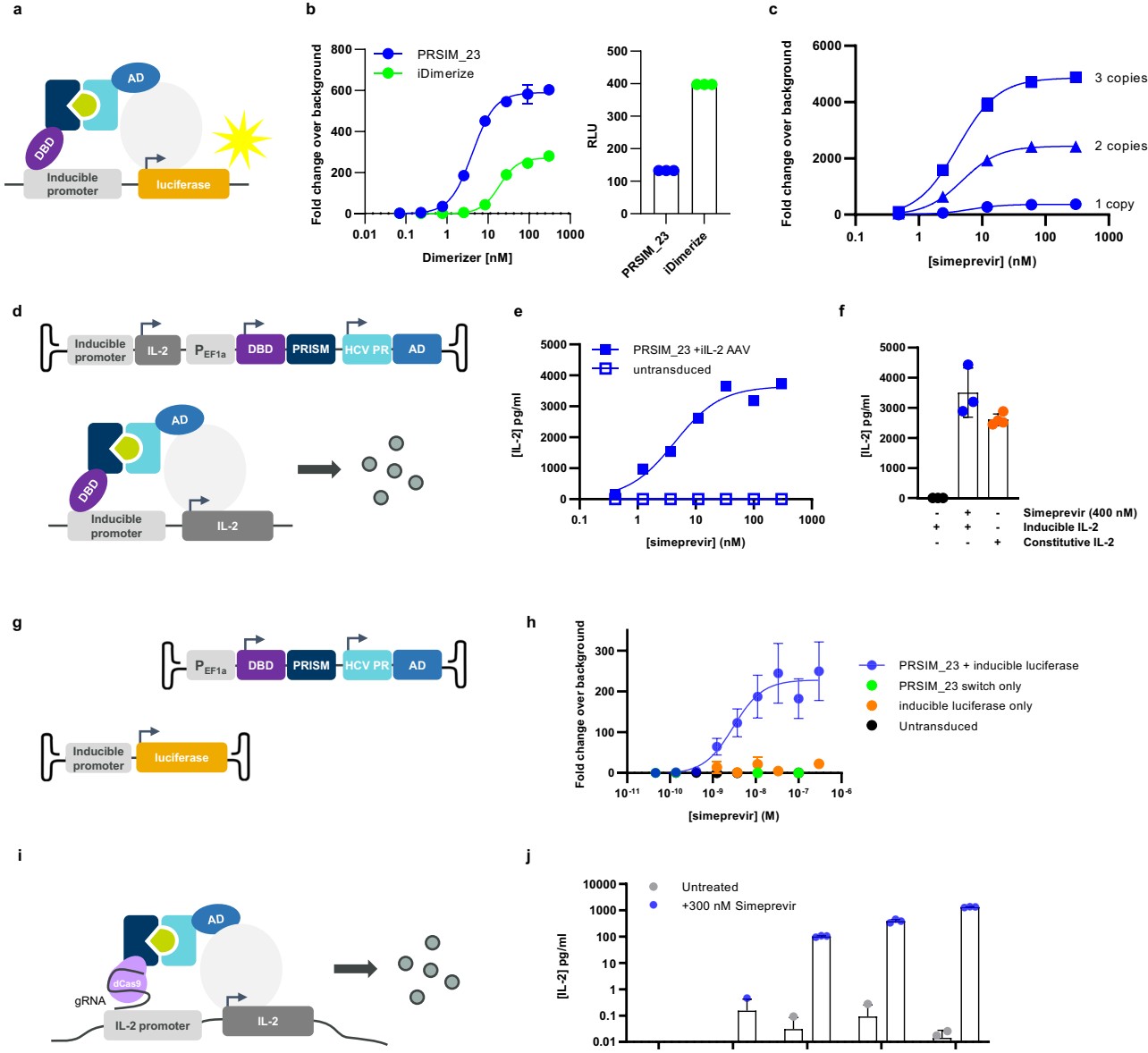

**Fig. 2 | The PRSIM-based CID module can regulate expression of exogenous and endogenous genes. a** Schematic of the simeprevir-inducible split transcription factor (STF) luciferase reporter assay. AD is activation domain and DBD is DNA-binding domain. **b** Left: the PRSIM_23-based STF system (blue) achieves a greater fold-induction of luciferase expression compared to the rapalog-induced iDimerize system (green). Right: A 3-fold lower level of baseline luciferase expression is observed from the PRSIM_23-based system (blue) than from the iDimerize system (green). RLU is relative luminescence units. **c** One (triangles) or two (squares) additional copies of PRSIM_23 fused to the AD lead to greater fold-induction of luciferase gene expression compared to a single copy (circles). **d** Schematic of AAV delivery of inducible IL-2 to cells. The PRSIM_23 CID STF and an inducible IL-2 transgene (iIL-2) are packaged into a single AAV particle. **e** Simeprevir-induced dose-dependent expression of IL-2 (squares) after transduction. Open squares indicate untransduced cells. **f** Levels of IL-2 expression induced by the PRSIM_23-based CID module (blue) were comparable to IL-2 levels from an AAV containing a constitutively active IL-2 transgene (orange). **g** Schematic of AAV vectors for delivery of the PRSIM_23 CID module-based STF and inducible luciferase transgene to cells in trans. **h** Simeprevir dose-dependent luciferase expression was observed after co-transduction (blue) but not after transduction of either AAV individually (green, orange) or in untransduced cells (black). **i** Schematic of the PRSIM_23 CID-based CRISPRa system used to regulate endogenous IL-2 expression. **j** Cells were co-transfected with an IL-2-targeting gRNA plasmid and a plasmid encoding the PRSIM_23 CID module-based STF. Expression of endogenous IL-2 was induced with 300 nM simeprevir (blue) or vehicle (grey) and either a single gRNA or two gRNAs in combination. For **b**, **c**, **e** and **h**, dose–response curves were fit to the data using 4-parameter nonlinear regression. Each data point represents the mean of two (**e**), three (**b**), (**f**; inducible IL-2), (**h**), (**j**), or four (**c**), (**f**; constitutive IL-2) independent replicates. Error bars in **b**, **c**, **f**, **h**, and **j** are s.e.m. Source data for **b**, **c**, **e**, **f**, **h**, and **j** are provided as a Source Data file.

and assessed the ability of the PRSIM-based CID module to regulate luciferase gene expression in the presence of increasing concentrations of simeprevir following transient transfection in HEK293 cells. The PRSIM_23-based construct demonstrated dose-dependent gene expression with an EC50 of 4.2 nM for simeprevir (Fig. 2b). In comparison, the iDimerize (FRB:FKBP12:rapalog) positive control has an EC50 of 18.8 nM. Furthermore, the baseline level of luciferase expression in the absence of dimerizer is ~3-fold higher for the FRB:FKBP12:rapalog system, indicating a greater degree of leakiness with this system than is observed for the PRSIM_23 CID module (Fig. 2b).

To investigate whether it would be possible to achieve greater levels of induction of gene expression via the split transcription factor system described above, we tested constructs containing one, two, or three copies of the Tn3 molecule PRSIM_23 as a fusion to the DBD. A graded level of luciferase expression was observed in HEK293 cells transiently expressing these components; each additional copy of the Tn3 molecule was able to elicit a significantly increased fold-change in luciferase expression level (2 copies, 2436-fold; 3 copies, 4862-fold), compared to the single copy (364-fold) (Fig. 2c). To confirm the utility of the PRSIM_23/HCV PR-based split transcription factor system for the regulation of other genes, we replaced the luciferase coding region with genes for an antibody (MEDI8852) or a luciferase-targeted shRNA (Supplementary Fig. 8) and observed dose-dependent induction of transgene expression in both cases.

Recombinant adeno-associated virus (rAAV) vectors are an effective means to deliver therapeutic transgenes into cells in vivo. Here we used AAV to deliver the DNA encoding an inducible IL-2 transgene to cells together with the PRSIM_23/HCV PR-based split transcription factor components described above. To demonstrate that the PRSIM_23/HCV PR-based split transcription factor and an inducible transgene can be delivered to cells in a single AAV particle, we generated AAV8 particles from an AAV vector encoding both the PRSIM_23/HCV PR-based transcription factor components and an inducible IL-2 transgene (Fig. 2d). After transduction of HEK293 cells with this AAV, we observed simeprevir-inducible dose-dependent regulation of IL-2 gene expression, with an EC50 of 4.3 nM for simeprevir and maximal levels of ~3500 pg/ml IL-2 observed (Fig. 2e). The level of IL-2 expression induced by the PRSIM_23/HCV PR-based CID module at the highest concentrations of simeprevir tested was comparable ($3506 \pm 817$ pg/ml) to that achieved from a control AAV8 encoding the IL-2 transgene under the control of a constitutive CAG promoter ($2606 \pm 189$ pg/ml) (Fig. 2f). To demonstrate that the PRSIM_23/HCV PR-based split transcription factor and an inducible transgene can function when delivered to cells in separate AAV particles, we generated AAV8 particles from an AAV vector encoding the PRSIM_23/HCV PR-based transcription factor components and separate AAV8 particles from a vector encoding an inducible luciferase transgene (Fig. 2g). After co-transduction of HEK293 cells with these two AAV, we observed simeprevir-inducible dose-dependent regulation of luciferase gene expression (Fig. 2h).

Having demonstrated that PRSIM-based CID modules can regulate the expression of an exogenous transgene delivered to cells by transfection or AAV transduction, we investigated whether the PRSIM-based CID system could also regulate the expression of endogenous genes. To demonstrate this, an inactive form of the *Streptococcus pyogenes* Cas9 enzyme (dCas9) and an activation domain (AD) consisting of a fusion of three transcriptional activators (VP64, p65, and Rta; VPR) were separately fused to the two protein components of the CID module (three copies of PRSIM_23 and HCV PR, respectively) (Fig. 2i). In HEK293 cells transiently expressing the PRSIM-regulated split dCas9/VPR cassette and two separate IL-2 targeted gRNAs (guide 1 and guide 8), the addition of simeprevir resulted in secretion of IL-2. Guide 8 induced a higher level of IL-2 production than guide 1, but when used together the two gRNAs induced IL-2 production to a

higher level than for both guides combined (Fig. 2j). Importantly, no IL-2 was detected in cells expressing the gRNA or PRSIM_23-dCas9 only or in those cells expressing a non-IL-2 targeting gRNA.

## PRSIM-based CID modules can regulate the activity of an apoptotic protein to control cell death in multiple cell types

The ability to remotely control cell therapies once they have been administered provides a safety net in the event of uncontrolled proliferation or unexpected toxicity. One way to control such cells is to endow them with a kill switch such that they can be removed at will once they have performed their function or pose a safety risk[26]. As such, a PRSIM-based, simeprevir-inducible Caspase 9-based kill switch was generated and tested in vitro. The homodimerization CARD domain of Caspase 9 was replaced with a fusion of the PRSIM_23 and HCV PR domains separated by short linkers. An active Caspase 9 homodimer can thus only be reconstituted by addition of simeprevir (Fig. 3a). Addition of simeprevir to HEK293, HCT116, and HT29 cells stably transduced with the kill switch construct induces rapid cell death upon addition of 10 nM simeprevir as observed microscopically (Fig. 3b). Active Caspase 9 activates downstream Caspase 3 by proteolytic cleavage and can be detected by cleavage of fluorogenic substrate Ac-DEVD-AMC. Caspase 3 activity was significantly upregulated in simeprevir-treated kill switch-transduced HEK293 cells (Fig. 3c) and HCT116 and HT29 cells (Fig. 3d). Simeprevir-induced Caspase 9 and Caspase 3 cleavage were confirmed by Western blot (Supplementary Fig. 9). Furthermore, cell killing via induction of apoptosis was simeprevir-dose-dependent, with EC50s of $1.64 \pm 0.56$ nM for HCT116 cells and $5.50 \pm 1.17$ nM for HT29 cells after 4 h, and $97.6 \pm 44.8$ pM for HCT116 cells and $342 \pm 106$ pM cells for HT29 after 48 h (Fig. 3e).

Caspase 9 can be inactivated by Akt kinase-mediated phosphorylation on Ser196. To mitigate the risk of this modification silencing the kill switch, we replaced the WT Caspase 9 with a variant containing the S196A mutation to prevent phosphorylation. The addition of 10 nM simeprevir to HEK293 cells stably transfected with the S196 kill switch induced rapid cell killing in a timeframe comparable to the WT kill switch (Fig. 3f). Activity of downstream Caspase 3 was significantly upregulated in both WT and S196A mutant kill switch cells compared to non-transduced cells; in the same assay, no significant differences between WT and S196A kill switch cells were detected (Fig. 3g). This result demonstrates that a kill switch mutant that prevents a cellular evasion mechanism is as active as the WT kill switch.

Pluripotent stem cells, including embryonic stem (ES) cells, have immense potential in regenerative medicine[27–29]. The ability of ES cells to differentiate into any cell type comes with a risk of teratoma formation in the event of undesired, uncontrolled differentiation of cells in patients[30]. To demonstrate that the PRSIM-based kill switch can eliminate this type of cell, stable cell lines containing the kill switch were made in the ES cell line Sa121. In kill switch-transduced ES cells, we observed cell killing with a dose–response to simeprevir. A high dose of simeprevir (1 µM) rapidly and efficiently eliminated up to 95% of cells within 4 h, as measured by cell confluency, with an onset of ~15 min (Fig. 4a). Lower doses of simeprevir initiated cell killing with a delayed onset; 100 nM of simeprevir was able to induce ~90% cell killing within 4 h, whereas at 10 nM simeprevir, maximal cell killing was not reached within the 4 h timeframe of the experiment. In contrast, WT Sa121 cells did not respond to treatment with simeprevir.

T cell-based therapies, including tumor-infiltrating lymphocytes, CAR T cells, and T-cell receptor (TCR)-T cells, are being widely explored to treat cancer[31]. To investigate whether the PRSIM_23-based kill switch is effective in T cells, we used a CRISPR/Cas9 knock-out/knock-in (KO/KI) strategy to engineer the PRSIM_23 CID-based kill switch into the *TRAC* locus in place of the endogenous T-cell receptor

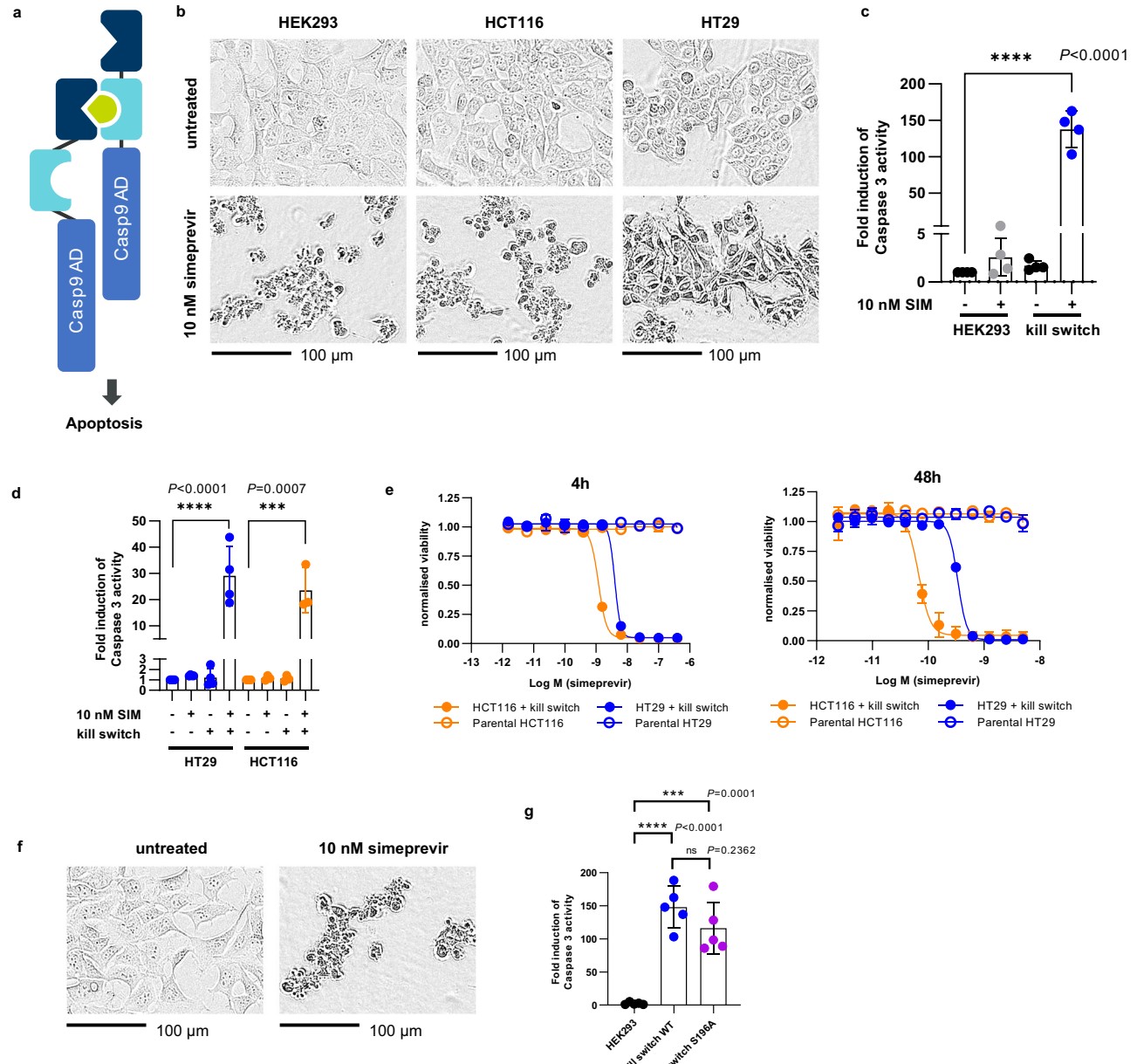

**Fig. 3 | The PRSIM-based CID module can regulate a Caspase 9-based kill switch in vitro. a** Schematic of the Caspase 9-based kill switch. The PRSIM_23-based CID protein components are fused to a dimerization-deficient Caspase 9 activation domain (AD). **b** Phase contrast images of HEK293 (left), HCT116 (middle), and HT29 (right) cells stably transduced with the wild-type kill switch show rapid cell death upon treatment with 10 nM simeprevir (bottom) but not in untreated cells (top). Images were acquired after 60 min (HEK293, HCT116) or 90 min (HT29). **c** Caspase 3 activity is induced in wild-type kill switch-transduced HEK293 cells after treatment with 10 nM simeprevir (SIM; blue) relative to untransduced untreated HEK293 cells (grey). Each data point represents the mean ± s.d. of four independent experiments. **d** Caspase 3 activity is induced in wild-type kill switch-transduced HCT116 (orange) or HT29 cells (blue) after treatment with 10 nM simeprevir relative to untransduced, untreated parental cells. Each data point represents the mean ± s.d. of four independent experiments. **e** Simeprevir-dose-dependent cell viability

4 hrs (left) and 48 hrs (right) post simeprevir dosing in HCT116 (closed orange circles) or HT29 cells (closed blue circles) transduced with the kill switch. No cell killing is observed in untransduced parental cells (open circles). Each data point represents the mean of three independent replicates ± s.d. Data were fit to a dose-response curve using 4-parameter nonlinear regression. **f** Phase contrast images of HEK293 cells stably transduced with the kill switch S196A mutant show rapid cell death upon treatment with 10 nM simeprevir (bottom) but not in untreated cells (top). **g** Caspase 3 activity is induced by 10 nM simeprevir to comparable levels in the wild-type (WT; blue) and S196A mutant (purple) kill switch relative to treated untransduced HEK293 cells. Each data point represents the mean of five independent experiments ± s.d. For **c**, **d**, and **g**, data was analyzed using one-way ANOVA with Dunnett's test for multiple comparisons. For **b**, **e**, and **f**, a representative of $n = 4$ independent experiments is shown. Source data for **c**–**e**, and **g** are provided as a Source Data file.

(Supplementary Fig. 10) in primary human CD8 + T lymphocytes from three different donors. After treatment of the kill switch-containing T cells with 10 nM or 10 µM simeprevir, we observe the disappearance of T cells as measured by flow cytometry, compared to untreated cells (Fig. 4b, Supplementary Fig. 10), with >90% of T cells eliminated after 21 days after the 10 µM dose.

## The PRSIM-based CID system can induce cell killing in a xenograft mouse model

We next investigated whether the simeprevir-inducible PRSIM_23 CID module incorporated into the Caspase 9-based kill switch could be used to regulate apoptosis in vivo, in the context of a tumor-bearing mouse model. Clonal HT29 cells that had been transduced with a

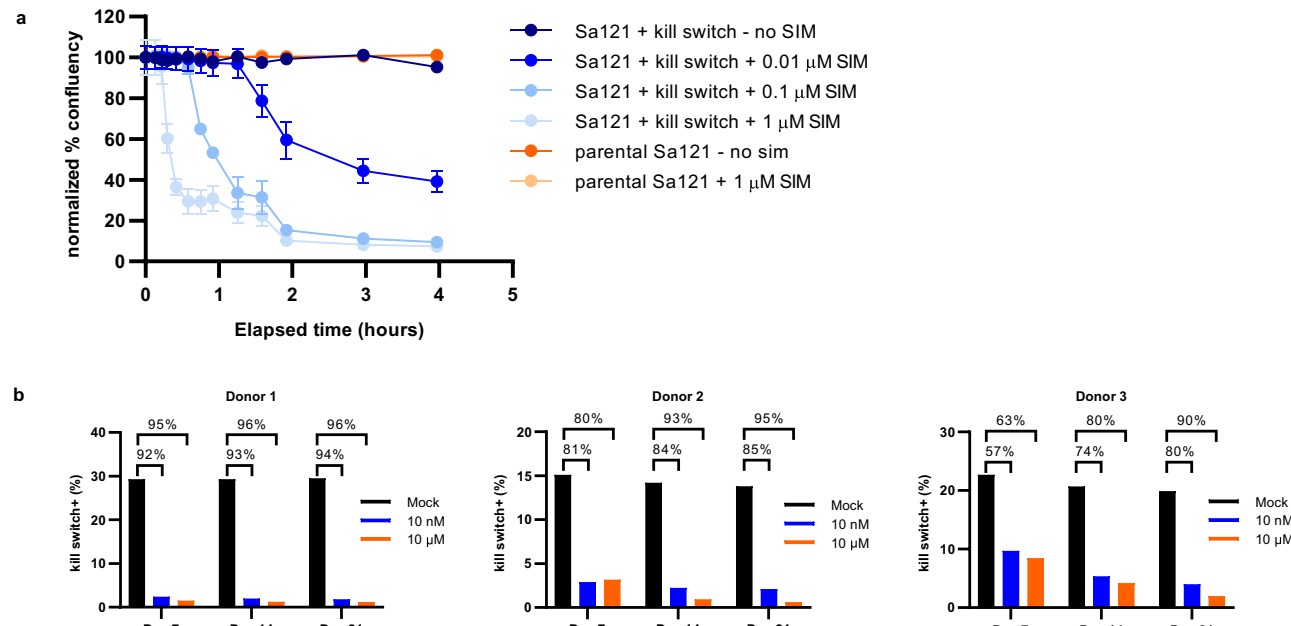

**Fig. 4 | The PRSIM-based CID module can be used to regulate a Caspase 9-based kill switch in therapeutically relevant cells. a** Simeprevir (SIM)-induced dose-dependent cell killing up to 4 hrs post simeprevir dosing in kill switch-transduced Sa121 hESCs (blue circles). No killing is observed in parental Sa121 hESCs (orange circles) or without simeprevir stimulation. Each data point represents the mean of three independent replicates ± s.d. **b** Functionality of the kill switch engineered into primary T cells was determined by flow cytometry, measuring the disappearance of live kill switch+ (CD19+) cells following 10 nM (blue) or 10 μM simeprevir (orange) treatment for 7, 14, or 21 days. Untreated cells are in black. Data is presented for n of 3 human donors with percentage killing denoted above each bar. Source data are provided as a Source Data file.

lentivirus expressing the simeprevir-inducible kill switch were implanted into SCID mice, and 200 mg/kg simeprevir was dosed orally when the average group tumor size reached 250 mm³ or 500 mm³ (22 or 34 days post-implantation, respectively) (Fig. 5a). The 200 mg/kg dose of simeprevir was chosen based on PK studies with simeprevir (Supplementary Fig. 11) which showed that this dose gave simeprevir exposure levels over the EC50 value measured in vitro for induction of the kill switch. Remarkably, we observed complete regression of the tumors in all mice, regardless of tumor size at dosing, after a single dose of simeprevir, with tumors remaining undetectable up to three weeks after dosing (Fig. 5b). In contrast, tumors in untreated mice continued to expand, and all mice had to be culled for welfare reasons by day 42 of the study. Importantly, body weight measurements of individual mice indicated no signs of toxicity due to dosing with simeprevir (Supplementary Fig. 12).

## Discussion

The identification of CID modules is of significant interest in the field of synthetic biology, as these modules can be used to engineer any cellular pathway dependent on protein proximity. Here we have described a CID module based on the clinically approved, well-tolerated antiviral simeprevir, and a catalytically inactive variant of its protein target, the HCV NS3/4A protease (S139A). We demonstrate that the PRSIM_23 CID module can be used to regulate gene expression in cells, of both exogenous and endogenous genes, via simeprevir-dependent reconstitution of a split transcription factor. We also show that the PRSIM_23 CID module can be incorporated into a Caspase 9-based kill switch, enabling rapid induction of apoptosis in response to simeprevir, both in vitro, in multiple therapeutically relevant cell types, and in vivo in a xenograft tumor model.

We designed the PRSIM_23 CID system with a particular focus on the potential for using this module to engineer cellular processes in therapeutic applications. The choice of the protein-small molecule pair of HCV PR and simeprevir was guided by this consideration.

Simeprevir has a clinically validated safety and pharmacokinetic profile suitable for once-daily and (if required) chronic dosing. The natural target of HCV PR is the viral polyprotein, rather than any human protein, and the catalytically inactive S139A variant used here should eliminate any residual off-target protease activity against human proteins. Thus, this protein-small molecule pair, in contrast to CID modules which are comprised of one or more human protein components, should be completely orthogonal to human biological proteins and pathways. Furthermore, as we envision that the PRSIM_23 CID module will be used intracellularly, we identified complex-specific binding partners for the HCV PR:simeprevir complex from a library of Tn3 variants with diversified CDR-like loops. This scaffold is small, monomeric, stable, and lacks Cys residues, and thus is well suited for intracellular applications, unlike scFvs which contain conserved disulfide bonds that are important for stability but incompatible with the reducing environment of the cytoplasm, and also may be prone to aggregation[32].

The PRSIM_23 CID system generated here with these design considerations in mind thus offers potential advantages over other previously described CID systems. For example, the approach used by Glasgow et al, in which ligand-binding sites are engineered into existing protein-protein interfaces, is potentially generalizable, but to date, it has only been applied to engineering CIDs as sensors for the toxic molecule FPP which would not be suitable for clinical use. The AbCID system[9] is based on the human anti-apoptotic protein Bcl-XL and the Bcl-XL inhibitor ABT-737, and the LITE system[10] is based on another related anti-apoptotic protein, BCL-2, and the approved small molecule inhibitor venetoclax. Thus, one potential limitation of these systems is that the expression of CID components has the potential to perturb endogenous apoptotic pathways. Moreover, the use of small molecule inducers with endogenous targets will require careful titration in order to induce heterodimerization of the CID module without on-target but undesired inhibition of the endogenous human protein, which will be challenging in a clinical setting. Akin to the PRSIM_23 CID

**a**

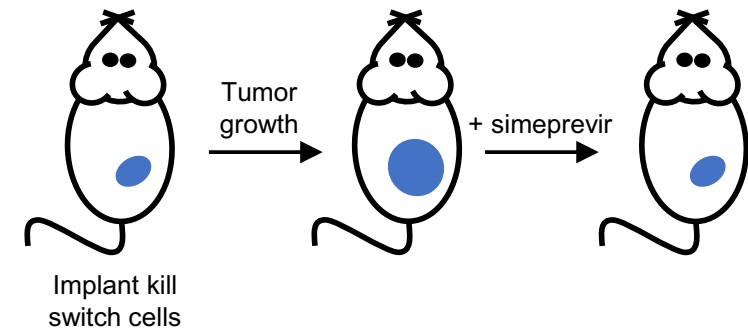

**b**

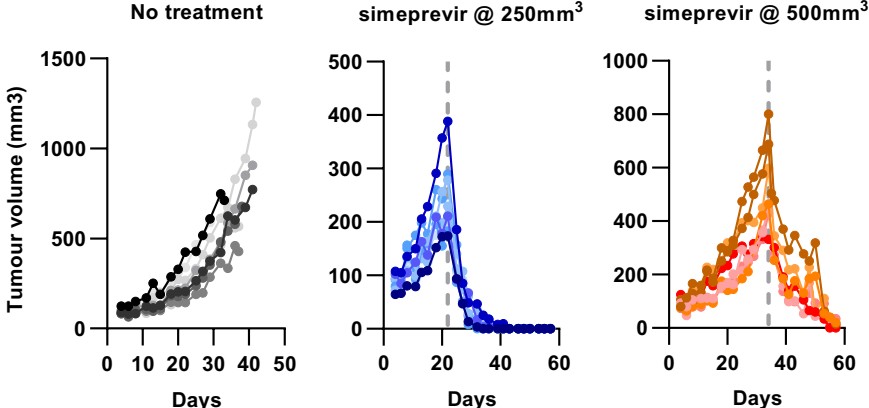

**Fig. 5 | The PRSIM-based CID can be used to regulate a Caspase 9-based kill switch in vivo. a** Schematic of mouse model. **b** Complete tumor regression is induced by a single 200 mg/kg dose of simeprevir in a kill switch-transduced HT29 xenograft mouse model. The dashed line represents the day of simeprevir dosing. Each line represents an individual mouse. Black circles are mice treated with vehicle only, blue circles are mice treated with simeprevir after the average group tumor size reached 250 mm³, and orange circles are mice treated with simeprevir after the average group tumor size reached 500 mm³. Source data for **b** are provided as a Source Data file.

system, the PROCISiR system[12] is based on the NS3/4A protease from hepatitis C virus (HCV) and clinically approved HCV PR inhibitors (grazoprevir and danoprevir). However, in the case of PROCISiR, the specific HCV PR-antiviral complex binding partners are de novo designed proteins requiring both computational design and optimization by yeast display in order to generate specific, high-affinity complex binders. In contrast, we identified the Tn3 component of the PRSIM_23 CID module using phage display, a robust and widely used display platform, with no further optimization required. Furthermore, the Tn3 scaffold used here is derived from human tenascin C, and thus has a low risk of immunogenicity, whereas the immunogenicity risk of the de novo-designed proteins in PROCISiR is unknown.

The strategy used here to generate the PRSIM_23 CID module could be extended to identify complementary or orthogonal CID modules. A number of clinically approved HCV PR inhibitors are available and could be used in combination with simeprevir to create a multi-input control system. As described here, phage display enables rapid identification of specific Tn3 binding partners for each HCV PR:antiviral complex which would be required to generate such a multi-input system. Similarly, phage display could also be used to generate CID modules which are based on approved antivirals that target other viral proteins instead of HCV PR. Such orthogonal CID modules could then be used in combination with the PRSIM_23 CID-based kill switch or split transcription factor to regulate additional cellular processes.

One potential risk in using the HCV PR protein as a component of the PRSIM_23 CID module is immunogenicity, in particular induction of CD8 + T cell responses which would lead to the elimination of cells expressing the PRSIM_23 CID module intracellularly. Immunogenicity is likely to be influenced by a number of factors, dependent on the particular application—for example, for AAV delivery of transgenes, immune response to transgenes is known to be influenced by dose, serotype, and target tissue, among other factors, and can potentially be mitigated by vector modifications (e.g., CpG depletion)[33]. In alternative applications, such as in the context of allogeneic cell therapies, where cells are typically engineered to remove the components required for antigen presentation, the risk of immunogenicity of the HCV PR component of the PRSIM_23 CID module will be low. In applications where immunogenicity is deemed a significant risk, the large body of literature available describing immunogenic epitopes in HCV PR resulting from efforts to design vaccines against HCV could be used to guide deimmunization strategies to remove T-cell epitopes from the protein sequence.

We have demonstrated that the PRSIM_23 CID system can be used to regulate the expression of exogenous genes delivered via AAV. Due to packaging limitations of AAV, the delivery in a single AAV particle of both the PRSIM_23 CID module—split transcription factor construct and the inducible transgene will be limited to transgenes <~550 bp (e.g., cytokines, shRNAs) when the CID module contains three copies of the Tn3 component. However, it may be possible to reduce the copy

number of PRSIM_23 in the split transcription factor from 3 copies, as used here, to 2 or even 1 copy, depending on the level of transgene induction required for a particular therapeutic application. In addition, as we have shown here, the PRSIM switch-split transcription factor and the inducible transgene can be delivered in trans, extending the size of potential transgenes to ~3.6 kb. In this respect, the PRSIM_23 has an advantage over the AbCID and LITE systems, as the gene encoding the single-chain Fab used in AbCID, which contains only one binding moiety, is 630 bp larger than that encoding three copies of the PRSIM_23 Tn3. Similarly, the synthetic reader protein monomers used in the PROCISiR system are twice the size of the Tn3 monomer in the PRSIM_23 system. For use of the PRSIM_23-based CID in the context of cell therapies – for example, to regulate the expression of a CAR or of a protein used for armoring – the size of the PRSIM_23 CID cassette is likely to be somewhat less of a limiting factor, as delivery of switch components and inducible transgenes is typically achieved using lentiviral transduction or via targeted integration using a DNA donor template.

Another potential application for the PRSIM_23-based switch, as demonstrated here, is for inducible regulation of endogenous genes, eliminating the need for delivery of an exogenous inducible gene. This broadens the applicability of the approach by enabling the regulation of genes of any size and mitigating the potential risks of expressing a protein at higher levels than is observed from its native promoter. Although the dCas9 and gRNA cassette used here for recognition of the endogenous promoter is large (thereby limiting options for delivery), this could be replaced by a smaller DNA-binding protein, such as a zinc finger, designed against the endogenous gene.

An inducible Caspase 9-based kill switch (iC9) triggered by the rapalog AP1903 is currently being investigated in the clinic as a safety switch for cell-based therapies[34–36]. Unlike simeprevir, AP1903 has not been approved for use in humans, though it was well tolerated in a Phase I study[37]. In preclinical studies, a single dose of the related CID compound AP20187 eliminates the majority of CD19 CAR T cells engineered to express the iC9 kill switch. Incorporation of the PRSIM_23 CID-based Caspase 9 kill switch into therapeutic cells should similarly enable rapid elimination of cells when required due to unexpected toxicity and/or stable disease remission. The activity of the PRSIM_23 CID-based Caspase 9 kill switch is also tuneable, enabling partial elimination of cells at low concentrations of simeprevir, which could be advantageous in a therapeutic context for controlling cytokine release syndrome while still retaining antitumor effects in the subset of cells remaining after apoptosis[38]. We have demonstrated that the PRSIM_23 CID-based kill switch is effective in both a human ES cell line and in human primary CD8 + T cells. Thus, the PRSIM_23 CID-based Caspase 9 kill switch has significant potential as a safety switch for a number of cell-based therapies.

## Methods
### Ethics statement
All human biological samples were commercially acquired from BioIVT. AstraZeneca has a governance framework and processes in place to ensure that commercial sources have appropriate patient consent and ethical approval for the collection of the samples for research purposes, including use by for-profit companies. The AstraZeneca BioBank in the UK is licensed by the Human Tissue Authority (Licence No. 12109) and has National Research Ethics Service Committee (NREC) approval as a Research Tissue Bank (RTB) (REC No. 22/NW/0102) which covers the use of the samples for this project.

The in vivo study was carried out in the United Kingdom and was conducted under a U.K. Home Office Project Licence in accordance with the U.K. Animals (Scientific Procedures) Act 1986 and in accordance with the EU Directive (EU 2010/63/EU). Welfare limits pertaining to tumor volume (no more than 1800 mm³ or an average diameter of

15 mm) or tumor condition (ulceration of the skin above the tumor) were not exceeded during this work.

### Constructs
The sequence used in the design of HCV NS3/4A PR constructs is derived from UniProt entry A8DG50 (hepatitis C virus subtype 1a genome polyprotein) and incorporates additional modifications from US patent US6800456. The protease domain corresponds to residues 1030-1206 of the polyprotein. A single chain consisting of an 11-residue peptide derived from the viral NS4A protein fused to the N-terminus of NS3 protease was used to create a fully folded and activated polypeptide. This sequence with an N-terminal hexahistidine (6His) tag and AviTag was purchased as a linear DNA string (GeneArt). In parallel, a DNA string encoding an equivalent sequence with the active site mutation S139A was ordered. The DNA strings were cloned into the pET-28a vector (for bacterial expression) using Gibson assembly.

Tn3s for bacterial expression were subcloned into the pET-16b vector after PCR amplification from the phagemid vector used for phage display.

To generate split transcription factor vectors, the DNA sequence encoding HCV NS3/4A PR (S139A) was purchased as a DNA string from GeneArt and cloned into the pHet-Act1-2 vector from the iDimerize regulated transcription system (Takara) as an N-terminal fusion partner to the activation domain with a flexible linker (TGGGGSGGGGS) between the fusion partners. Subsequently, sequences encoding one, two or three copies of a panel of the Tn3 molecules (termed HCV NS3/4A PR:simeprevir complex-specific binding (PRSIM) molecules) were purchased as DNA strings from GeneArt and were cloned using Gibson assembly into the HCV NS3/4A PR (S139A)-containing pHet-Act1-2 construct as a fusion partner to the DNA-binding domain (DBD).

AAV expression vectors were generated by subcloning specific promoter and transgene elements using appropriate restriction sites and/or In-Fusion cloning (Takara) into an intermediate vector derived from pAAV-CMV (Takara) in which the CMV promoter downstream of the 5′ ITR was removed and a WPRE element and SV40 polyA sequence were inserted upstream of the 3′ ITR. To generate AAV encoding an inducible luciferase transgene, the ZHFD1-luciferase cassette was amplified by PCR from pZFHD1-Luciferase from the iDimerize regulated transcription system (Takara) and cloned into the intermediate AAV vector. To generate AAV encoding constitutively expressed human interleukin-2 (IL-2), a gene encoding human IL-2 was subcloned downstream of a CAG promoter in the intermediate AAV vector. To generate AAV encoding the PRSIM_23 CID module in the context of the split transcription factor, a cassette encoding two fusion proteins (the ZFHD1 DNA-binding domain fused to 3 copies of PRSIM_23 and HCV NS3/4A PR (S139A) fused to the AD) separated by a P2A self-cleaving peptide was cloned downstream of a hybrid EF1alpha-HTLV-1 promoter in the intermediate AAV vector. To generate AAV encoding an inducible IL-2 transgene in addition to the PRSIM_23 CID module-split transcription factor, human IL-2 was cloned in place of the luciferase transgene in the pZFHD1-Luciferase vector, and the ZFHD1-IL-2 cassette was amplified by PCR and inserted immediately downstream of the 5′ ITR in the AAV vector encoding the PRSIM_23 CID module-split transcription factor construct.

For vectors for regulation of endogenous gene expression, the activation plasmid was generated consisting of two functional units; an activation domain (AD) fused to the HCV NS3/4A PR (S139A) and a dCas9 fused to three tandem copies of PRSIM_23. The sequences are preceded by a CMV promoter and separated by an internal ribosome entry site (IRES). A gRNA plasmid was generated by Golden Gate assembly, utilizing BsaI. The gRNA plasmid encodes the human U6 promoter, IL-2 target sequence(s) (GTTACATTAGCCCACACTT and/or ATTCTGGAAAAATATTATGG), and a scaffold RNA sequence to allow Cas9 binding.

For a generation of kill switch vectors, the sequence encoding a kill switch fusion protein of PRSIM_23, HCV NS3/4A PR, and ΔCARD-Caspase9 with a short GGGSG linker between the three fragments was purchased as a cloned gene in vector pcDNA3.1 from GeneArt (Life Technologies). The fusion protein was subcloned into EcoRI/SalI digested lentiviral vector pCDH-EF1α-MCS-(PGK-GFP-T2A-Puro) (Systems Bioscience) using Gibson assembly cloning. To introduce the Caspase 9 S196A mutation into the kill switch construct, a short DNA fragment containing GCC (A) instead of AGC (S) in position 196 was cloned into the kill switch as a ClaI/Not fragment replacing the WT sequence.

The sequences of all final constructs were verified via Sanger sequencing. Protein and guide RNA sequences are shown in Supplementary Table 4.

## Protein expression and purification

For expression of HCV NS3/4A PR, the pET-28a plasmids were transformed into BL21(DE3) *E. coli* cells and selected on plates containing kanamycin (50 µg/ml). For each expression, a single colony was used to inoculate a 5 ml 2× TY + 50 µg/ml kanamycin culture that was grown at 37 °C overnight. This culture was used to inoculate 500 ml TB Autoinduction medium (Formedium, supplemented with 10 ml/L glycerol and 100 µg/ml kanamycin) at 1:500 dilution. The culture was grown at 37 °C to an OD600 of 1.3–1.5 and then transferred to 20 °C for 20 h for expression to be induced. Cells were harvested by centrifugation and the pellets were stored at −80 °C.

For HCV NS3/4A PR purification, each bacterial pellet from 500 ml culture was thawed and resuspended in 50 ml lysis buffer (2× DPBS, 200 mM NaCl, pH 7.4). The cells were lysed using a probe sonicator and the lysate was clarified by centrifugation at $50,000 \times g$ for 40 min at 4 °C, and filtered with 0.22 µm bottle-top filtration devices. The filtered supernatant was loaded on a 5 ml HisTrap HP column (Cytiva) at a 5 ml/min flow rate. The column was washed with 100 ml wash buffer (2× DPBS, 200 mM additional NaCl, 20 mM imidazole, pH 7.4) and eluted with an imidazole gradient over 5 column volumes from 20–400 mM imidazole. Fractions were analyzed by SDS-PAGE and those that were enriched for the correct protein were pooled and buffer exchanged with a HiPrep 26/10 Desalting column (Cytiva) into lysis buffer (2× DPBS, 200 mM NaCl, pH 7.4). Desalted protein fractions were pooled, concentrated with a centrifugal concentration device, and purified on a HiLoad Superdex 75 26/600 pg column (Cytiva) equilibrated in 2× DPBS, 2 mM DTT, 10 µM ZnCl$_2$. Fractions were analyzed by SDS-PAGE and those that were >95% pure were pooled, had their concentration determined via UV absorbance, and were snap-frozen in liquid nitrogen prior to storage at −70 °C. Final sample purity was verified with RP-HPLC on an XBridge BEH300 C4 column (Waters).

Tn3s were expressed in *E. coli* from phagemid vectors[39] in the periplasm in TG1 *E. coli* or from the pET-16b vector in the cytoplasm in BL21 (DE3) *E. coli*. Following preparation of periplasmic extracts by osmotic shock or lysis in BugBuster plus Benzonase (EMD Millipore), respectively, Tn3s were purified to homogeneity using nickel-chelate chromatography followed by size exclusion to provide a monomeric protein in PBS (pH 6.5).

## Protein biotinylation

Purified HCV NS3/4A PR was biotinylated on its AviTag using an MBP-tagged BirA enzyme incubated for 2.5 h at 22 °C in the presence of ATP and biotin. Biotinylated protein was purified via size exclusion chromatography on a HiLoad Superdex 75 16/600 pg column (Cytiva) in 2× DPBS, 2 mM DTT, 1 µM ZnCl$_2$. Fractions were analyzed by SDS-PAGE and those containing the protease were pooled and the extent of biotinylation was confirmed by intact mass spectrometry on a Xevo G2-CS MS (Waters). Biotinylated protein was split into aliquots, snap-frozen in liquid nitrogen, and stored at −70 °C.

## Phage display selections

Tn3 sequences were isolated from phage display selections[39,40] using a Tn3 library developed as a FnIII alternative scaffold based on the third such module in human tenascin C[21–23]. Biotinylated HCV NS3/4A PR (S139A) was pre-incubated with a 50-fold molar excess of simeprevir (MedChemExpress) and captured on streptavidin-coated magnetic beads (Promega). In total, four rounds of phage display selection were performed, using decreasing concentrations of biotinylated HCV NS3/4A PR and simeprevir (Round 1: 250 nM biotinylated HCV NS3/4A PR (S139A) + 12.5 µM simeprevir; Round 2: 100 nM biotinylated HCV NS3/4A PR (S139A) + 5 µM simeprevir; Round 3: 25 nM biotinylated HCV NS3/4A PR (S139A) + 1.25 µM simeprevir; Round 4: 25 nM biotinylated HCV NS3/4A PR (S139A) + 1.25 µM simeprevir).

Prior to each selection, the phage pool was incubated with streptavidin beads alone to deplete the library of any binders to the streptavidin beads. In rounds 3 and 4, two selections were performed in parallel, one of which included an incubation step with biotinylated HCV NS3/4A PR (S139A) in the absence of simeprevir. The protease was removed by incubation with streptavidin beads and the remaining phage was then added to the biotinylated HCV NS3/4A PR (S139A) coated on streptavidin beads in the presence of simeprevir for the selection protocol. To remove non-specifically bound phage, the beads were washed three times with D-PBS (Sigma), and bound phage was eluted with trypsin. Eluted phage was used to infect mid-log phage cultures of *E. coli* TG1 cells and plated on agar plates (containing 100 µg/ml ampicillin and 2% (w/v) glucose).

## Phage ELISA

Individual phage clones from round 3 and round 4 were picked for DNA sequencing and screening for antigen binding by phage ELISA[41]. Briefly, individual TG1 colonies encoding phage clones were grown in 96-well plates at 37 °C shaking at 280 rpm to log phase in media containing 100 µg/ml ampicillin and 2% (w/v) glucose. Helper phage was then added to each well and the plates incubated at 37 °C for 1 h, shaking at 150 rpm. Plates were then centrifuged at $4415 \times g$ for 10 min at room temperature and the media was removed and replaced with media containing 100 µg/ml ampicillin and 50 µg/ml kanamycin. Plates were then incubated overnight at 25 °C, shaking at 280 rpm. The following day, phage preparations were blocked by adding an equal volume of 2× PBS containing 6% (w/v) skimmed milk powder (Marvel) to each well of the plate.

Biotinylated HCV NS3/4A PR (S139A) was used to coat 96-well streptavidin-coated plates at 5 µg/ml (1.875 µM) in the presence and absence of a threefold excess of simeprevir (5.6 µM). Coated plates were washed with PBS and blocked with PBS containing 3% (w/v) skimmed milk powder (Marvel) for one hour. Following this blocking step, the plate wells were washed three times with PBS, prior to adding the blocked phage. After 1 h incubation at room temperature, plates were washed three times with PBS/Tween 20 (0.1% v/v). Phage that bound specifically to the antigen-coated plate was detected using a 1:5000 dilution of the anti-M13 phage-HRP tagged antibody (Cytiva), followed by detection using 3,3′,5,5′-Tetramethylbenzidine (Sigma). The detection reaction was stopped using 0.5 M H$_2$SO$_4$ and plates were read using an EnVision fluorescent plate reader (PerkinElmer) at 450 nm.

## HTRF-binding screens

Tn3 PRSIM binding molecules that were selective for the HCV NS3/4A PR (S139A):simeprevir complex were identified in HTRF assays run in parallel to measure binding in the presence and absence of simeprevir. HCV NS3/4A PR (S139A) and serial dilutions of purified PRSIM binding molecules were prepared in assay buffer (PBS containing 0.4 M potassium fluoride and 0.1% BSA). Streptavidin cryptate (Cisbio) was pre-mixed with anti-FLAG XL665 in the assay buffer. For each binding assay, 2.5 µl of Tn3 sample was added to 2.5 µl HCV NS3/4A PR (S139A)

(final assay concentration = 5 nM) and 2.5 μl of pre-mixed detection reagents. Either 2.5 μl simeprevir (final assay concentration = 50 nM) or 2.5 μl of a DMSO blank were also added to each well. The background was defined using wells with zero sample addition. For the selectivity assays simeprevir was substituted with alternate HCV PR small molecule inhibitors (MedChemExpress). For competition assays, simeprevir was pre-incubated with HCV protease before being added to the assay as a single 2.5 μl addition. A further 2.5 μl addition of the competing small molecule was also made. Assay plates were incubated overnight at 4 °C, prior to reading the time-resolved fluorescence at 620 nm and 665 nm emission wavelengths using an EnVision plate reader (PerkinElmer). Data were analyzed by calculating % Delta $F$ values for each sample. Delta $F$ was determined according to the following equation:

$$\%\text{Delta } F = (((\text{sample 665nm/620nm ratio value}) \\ - (\text{background 665nm/620nm ratio value}))/ \\ (\text{background 665nm/620nm ratio value})) \times 100 \quad (1)$$

Selective binding molecules are defined as those Tn3 PRSIM binding molecules that bind to the HCV NS3/4A PR (S139A):simeprevir complex and exhibit no binding to HCV NS3/4A PR (S139A) alone.

## SPR-binding kinetics analysis

The affinity of Tn3 PRSIM binding molecules was measured using SPR on the Biacore 8 K (Cytiva) at 25 °C using Biacore 8 K Control Software. Tn3s were covalently immobilized to a CM5 chip surface using standard amine coupling techniques at a concentration of 1 μg/ml in 10 mM sodium acetate pH 4.5.

The HCV NS3/4A PR (S139A), or bovine serum albumin (BSA) control, was diluted 1:4 (1.25–20 nM) ± 10 nM simeprevir in 10 mM HEPES pH 7.4, 150 mM NaCl, 0.05% Surfactant P20, 0.01% DMSO, ensuring constant simeprevir and DMSO concentrations. Alternatively, the HCV NS3/4A PR (S139A) was diluted 1:1 (11.7–3000 nM) in the absence of simeprevir. The samples flowed over the chip at 50 μl/min using single-cycle kinetics, with 2 min association and 10 min dissociation. The chip surface was regenerated with two 20 s pulses of 10 mM Glycine-HCl pH 3.0. The final sensorgrams were analyzed using the Biacore Insight Evaluation Software (Cytiva) and the apparent affinity constant $K_{D,app}$ was determined using a 1:1 binding model. To estimate the effectiveness of the complex formation and to compare the Tn3s, the theoretical $R_{max}$ (maximal feasible binding) was calculated for each curve based on the immobilized level of the Tn3s ($R_{ligand}$) and used to calculate % of theoretical $R_{max}$:

$$\text{Theoretical } R_{max} = \frac{R_{ligand(Tn3)} \times MW_{analyte(HCV\ NS3/4A\ PR(S139A))} \times Valency_{ligand(Tn3)}}{MW_{ligand(Tn3)}} (RU) \quad (2)$$

$$\% \text{ of theoretical } R_{max} = \frac{R_{max}}{\text{Theoretical } R_{max}} \times 100 (\%) \quad (3)$$

To measure the effect of simeprevir concentration on the formation of the HCV NS3/4A PR (S139A):PRSIM-binding molecule complex, simeprevir was diluted 1:2 (0.0152–300 nM) in a base buffer composed of 10 mM HEPES pH 7.4, 150 mM NaCl, 0.05% Surfactant P20 and 0.3% DMSO, in the presence of a constant 40 nM HCV NS3/4A PR (S139A) concentration. The samples flowed over the chip at 50 μl/min using multi-cycle kinetics, with 4 min association and 10 min dissociation. Titration curves for the induction of HCV NS3/4A PR (S139A):PRSIM dimerization by simeprevir were generated. The response for each simeprevir concentration at 225 sec (15 sec before the end of the association) was normalized as a percentage of the response for 300 nM simeprevir at 225 sec and plotted against the simeprevir concentration using GraphPad Prism. Each data point

represents the mean of three independent experiments ± s.e.m. The $EC_{50}$ reported was calculated using a nonlinear regression curve fit.

## Transcriptional regulation assays

Transcriptional regulation assays were performed in adherent HEK293 cells (ATCC CRL-1573) cultured in 384-well plates. Cells enzymatically dissociated from a tissue-culture flask were counted and plated at $7.5 \times 10^3$ cells/well in a 384-well plate. The plates were incubated overnight at 37 °C with 5% $CO_2$ to allow the cells to adhere. On day 2, the cells were co-transfected with a pHet-Act1-2 plasmid (containing the FRB:FKBP12 control fusion proteins (Clontech) or the HCV NS3/4A PR (S139A):PRSIM fusion proteins) and a pZFHD1 plasmid encoding luciferase using Lipofectamine LTX (ThermoFisher). On day 3, wells were treated with different concentrations of either A/C heterodimerizer (for the FRB:FKBP12 control), simeprevir, or vehicle control, and 24 h later luminescence was quantified with an EnVision plate reader (PerkinElmer) immediately following addition of SteadyGlo luciferase substrate (Promega). Alternatively, reverse transfections were carried out on Day 1, addition of dimerizer on Day 2 and luminescence quantified 24 h later on Day 3. Luminescence readings were converted into fold-change by dividing the signal in the presence of simeprevir by that in the absence of simeprevir.

Endogenous transcriptional regulation assays were performed in adherent HEK293 cells cultured in 96-well plates. Cells enzymatically dissociated from a tissue-culture flask were counted and plated at $2.5 \times 10^4$ cells/well. The plates were incubated overnight at 37 °C with 5% $CO_2$ to allow the cells to adhere. On day 2, the cells were co-transfected with the activation and gRNA plasmids using Lipofectamine 3000 (ThermoFisher), with a gRNA:activation plasmid DNA ratio of 2:1. On day 3, wells were incubated with 300 nM simeprevir or with vehicle control. 72 h post-treatment (day 6), the cell supernatant was harvested and IL-2 was quantified using a V-PLEX Human IL-2 Kit (Meso Scale Discovery), as per the manufacturer's protocol. Data were analyzed using Discovery Workbench.

## AAV production

Recombinant AAV (rAAV) was produced by triple-transfection of 40 T175 cm² flasks containing HEK293 T-17 (ATCC CRL-11268) cells at 80% confluency using a standard helper-free approach. Briefly, each flask was transfected with 15 μg of a helper plasmid (a plasmid containing adenoviral E2A and E4), 7.5 μg of the AAV ITR-bearing, and transgene-encoding plasmid and 7.5 μg of the AAV capsid plasmid (containing the AAV8 capsid and the corresponding Rep genes) using 90 μg of 40 kDa linear polyethylenimine. Five days after transfection, media was collected from all the flasks, treated with 2000 units of Benzonase nuclease, and incubated at 37 °C for 1 hr. The media was then filtered through a 0.22 μm filter and concentrated to a volume of 80 ml using tangential flow filtration (TFF). This volume was further concentrated and buffer exchanged with PBS using an Amicon-15ml-100 kDa filter before loading onto a stepwise iodixanol gradient (15%/25%/40%/60%) and spinning at $490,573 \times g$ on an ultracentrifuge in a Ti70 rotor for 1.5 hrs at 18 °C. Fractions were taken from the ultraclear centrifuge tubes by piercing the tube with a 19 gauge syringe in the 60% layer below the clear band representing the virus and the purity of each fraction was assessed by SDS-PAGE of each fraction and subsequent Sypro Ruby analysis. Pure fractions were combined, the buffer was exchanged with PBS in an Amicon-15ml-100 kDa filter and concentrated to a final volume of 150 μl and stored at −80 °C in aliquots to avoid any repeated freeze/thaws. The viruses were titered using digital-droplet PCR and a TaqMan probe specific to the ITRs. Typical titers ranged from $1–3 \times 10^{13}$ genome copies (GC)/ml.

## Transcriptional regulation assays with AAV delivery

All rAAV transduction assays were performed in adherent HEK293 cells cultured in 96-well plates. Cells enzymatically dissociated from the

tissue-culture flask were counted and plated at $2.5 \times 10^4$ cells/well in a 96-well plate. The plates were incubated overnight at 37 °C with 5% $CO_2$ to allow the cells to adhere. On Day 2, the cells were transduced with $2.5-5 \times 10^9$ GC/ml (corresponding to a multiplicity of infection (MOI) of $1-2 \times 10^5$) of the relevant rAAV. After incubation for 48–72 h, the cells were treated with different concentrations of simeprevir or with vehicle control and incubated for a further 24 h. For luminescence assays, SteadyGlo luciferase substrate (Promega) was added and luminescence was quantified with an EnVision plate reader (PerkinElmer). Luminescence readings were converted into fold-change by dividing the signal in the presence of simeprevir by that in the absence of simeprevir. For IL-2 assays, the supernatant was harvested and IL-2 was quantified using a V-PLEX Human IL-2 Kit (Meso Scale Discovery) following the manufacturer's protocol. Data were analyzed using Discovery Workbench.

## Kill switch cell line generation

Lentiviral particles encoding the WT or S196A kill switch were generated using the pPACKH1 HIV lentiviral packaging kit (Systems Bioscience), according to manufacturer's instructions. HEK293 cells were transduced for 24 h in the presence of 8 µg/ml polybrene after which cells were changed into fresh growth medium (DMEM + 10% fetal bovine serum + 1% Non-essential amino acids). 24 h later, transduced cells were selected by the addition of 1 µg/ml puromycin for 5 days. Before functional testing, transduced cell pools were FACS sorted based on GFP fluorescence to isolate high-expressing cell line pools and single-cell clones. HCT116 (ATCC CCL-247) and HT29 (ECACC 85061109) transduced cells were generated following the same protocol with the exception of using McCoy's 5 A medium + 10% fetal bovine serum as a growth medium, supplemented with 2 µg/ml puromycin for selection of transduced cells.

hESCs (Human Embryonic Stem Cell Line 121, Cellartis TakaraBio Y00020) were seeded at 35,000 cells/cm² in Cellartis DEF-CS 500 Culture System (TakaraBio) and were transduced 24 h after seeding. The cells were transduced for 24 h in the presence of 5 µg/ml polybrene, after which the medium was replaced with a fresh complete growth medium. 24 h later, the selection was initiated using 1ug/ml puromycin. Selection was maintained throughout culture except after passage, and the medium changed daily.

Primary human T cells (CD8+) were isolated from healthy donor Leukopak (BioIVT) in accordance with Human Tissue Act (HTA) regulations. Peripheral blood mononuclear cells (PBMCs) were isolated from whole blood by Ficol centrifugation using SepMate tubes (StemCell). T cells (CD8+) were subsequently isolated using negative selection isolation kits (StemCell) following the manufacturer's instructions. T cells (CD8+) were cultured in AIM-V Medium (Thermofisher Scientific) supplemented with 5% heat-inactivated human AB serum (Merck) & 300 IU/ml IL-2 (StemCell). T cells were seeded at $2 \times 10^6$ cells/ml (typically $10 \times 10^6$ cells per well of a six-well plate) and activated with ImmunoCult T-Cell activator (StemCell) following manufacturer's instructions. Additionally, cells were cultured in the presence of DNAPK inhibitor (DNAPKi, AZ13880164, 1 µM final concentration) for 1 h pre and 24 h post-editing to further promote Knock-In (KI).

T cells were engineered by CRISPR/ Cas9 KI at the human *TRAC* locus[42] to express the WT kill switch linked via P2A to a GPI-anchored surface expressed CD19 (1–333aa) under the control of a minEF1α promoter. RiboNuclearProteins (RNPs) were produced by mixing 125 pmol of synthetic sgRNA (CAGGGTTCTGGATATCTGT) (Merck; resuspended in TRIS-EDTA (pH 7.5)) with 50 pmol recombinant *sp*Cas9 protein (AZ in-house) at room temperature for 30 min (2.5:1 sgRNA to *sp*Cas9 molar ratio). Immediately prior to electroporation, T cells were pelleted by centrifugation at $300 \times g$ for 5 min, washed once in PBS, and resuspended to $1 \times 10^5$ cells/µl in Lonza electroporation buffer P3. $4 \times 10^6$ T cells were mixed in a 96 v-bottomed plate with the CRISPR/ Cas9 RNP and dsDNA HDRT (3 µg per million cells) in 100 µl total

volume before being electroporated using the Lonza 4D electroporation system, pulse code EH-100. Immediately after electroporation cells were carefully transferred to final culture vessels containing pre-warmed media supplemented with DNAPKi (AZ13880164, 1 µM final concentration). AAV6 (Vector Builder; MOI 200,000) encoding the HDRT for KI was added to cultures 2 h post electroporation. 24 h post electroporation medium was changed in order to remove inhibitors and virus, and 7 days later KI frequencies were determined via measurement of CD19 surface expression by flow cytometry using the BD LSR-Fortessa Cell Analyzer (BD Bioscience).

## Kill switch cell viability and Caspase 3 functional assay

HEK293, HCT116, or HT29 cells were plated at a concentration of 1e4 per well onto collagen-coated 96-well plates and 24 h later treated with 10 nM simeprevir. Phase contrast images were acquired at 10 min intervals using ×10 or ×20 objectives on an Incucyte S3 (EssenBioscience). hESCs (kill switch+ line and parental line) were plated onto 96-well plates at 30,000 cells/cm². 48 h after plating, cells were treated with three different concentrations of simeprevir (10 nM, 100 nM, 1 µM). Phase contrast images were acquired at various timepoints up to 4 h after the start of simeprevir treatment, using the ×10 objective on an Incucyte S3 (EssenBioscience) and cell confluency was determined using Incucyte analysis software. Results were normalized to confluency of the relevant cell line at time zero. For the Caspase 3 assay, cells were plated in duplicate onto 6-well tissue-culture treated plates at a concentration of 5e5 per well. After 24 h, one of the duplicate wells was treated with 10 nM simeprevir for 3 h. Cell lysates were analyzed in triplicate using a Caspase 3 assay from BD Biosciences according to manufacturer's instruction with the modification that total protein input was normalized to 50 µg by BCA assay (Thermo Fisher Scientific). Fluorescence was determined on an EnVision plate reader (PerkinElmer), with excitation at 380 nm and emission measured at 430 nm. For quantification, RFU (relative fluorescence unit) for wells that contained only the assay substrate were subtracted from all RFU derived from assay samples. Results were normalized to non-transduced cells that had not been treated with simeprevir. Analysis was performed in Prism (GraphPad) using a One-Way ANOVA followed by multiple comparisons. For the determination of EC50, HCT116 or HT29 cells were plated at 1e3 (48 h) or 2e3 (4 h) per well in 384-well plates in triplicate. 24 h later, the cells were treated with a twofold (48 h; from 5 nM) or fourfold (4 h; from 400 nM) serial dilution of simeprevir. Cell viability was determined by CellTiterGlo2.0 assay (Promega) according to manufacturer's instructions after 4 h incubation for cells plated at 2e3 or 48 h for cells plated at 1e3. Cell viability was normalized to non-treated cells and EC50 values were determined in PRISM (GraphPad) using log (inhibitor) vs response variable slope.

$1 \times 10^6$ kill switch+ KI cells were cultured per well of a 48-well plate in the presence or absence of 10 nM or 10 µM simeprevir. At days 7, 14 & 21 post-simeprevir treatment the disappearance (death) of live TCR-kill switch+ (CD19+) cells was determined by flow cytometry; $1.5 \times 10^5$ cells were pelleted at $300 \times g$ for 5 min, washed in PBS, and subsequently stained using the Zombie Violet™ Fixable Viability Kit (Biolegend) for 20 mins at RT in the dark. Subsequently, cells were washed in FACS buffer (PBS, 1% BSA) and stained using PE anti-human α/β T-Cell Receptor Antibody (Biolegend) and APC anti-human CD19 Antibody (Biolegend) both at 1 in 100 dilution for 30 mins at 4 degrees. Cells were acquired using the BD LSR-Fortessa Cell Analyzer (BD Bioscience) and data files were analyzed using FlowJo.

## HT29 xenograft study

Female SCID mice (C.B-17/IcrHan®Hsd-Prkdcscid strain) were bred at Envigo UK, supplied at 8 weeks of age, and ~18 g. Sex was not considered in the study design; mouse sex was not expected to impact the results of the study as this study measures the activation of the kill switch within the human cells that comprise the human tumor

xenograft. Furthermore, the kill switch is equally effective in HCT116 cells (female) and HT29 cells (male). All mice were housed in sterile and standardized environmental conditions (temperature between 19 °C and 23 °C, humidity between 45% and 65%, 12 h light/dark cycle). Mice received autoclaved food and bedding and drinking water ad libitum. At day 0 mice were shaved on the right flank and anesthetized with isoflurane. 100 μl of a solution containing $1 \times 10^7$ HT29 PRSIM_23-HCV PR NS3/4A (S139A)-Casp9 clone 9 cells in 50% growth factor–reduced Matrigel (Corning) was injected subcutaneously into the right flank of each mouse. Cells did not undergo any in vivo passaging and were maintained under limited passage from original stocks (typically under 5). Tumor volume was measured 3 times per week with callipers using the formula volume = $(\pi/6)*(width^2 \times length)$. Group sizes were powered on the basis of existing tumor growth data. Animals were randomized to treatment groups (8 mice per group) on the basis of tumor volume before the first treatment. SCID mice were treated orally with a single dose of simeprevir at 200 mg/kg when the average HT29 tumor volume reached thresholds of 250 mm³ (day 22) or 500 mm³ (day 34). One group was untreated as a negative control. Simeprevir in polyvinylpyrrolidone (PVP K30) was reconstituted in water just prior to being dosed orally (PO) at 20 mg/ml and 10 ml/kg (200 mg/kg). Mice were humanely killed before reaching welfare limits pertaining to tumor volume (no more than 1800 mm³ or average diameter of 15 mm) or tumor condition (ulceration of the skin above the tumor). This maximum burden was not exceeded during this work. For the mice where tumors regressed, the study was terminated at day 57.

## Data availability

Source data are provided with this paper.

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

## Acknowledgements

We thank the Biologics Expression Team for Tn3 expression & purification, James Button for inducible Medi8852 experiments, Silvia Sonzini for ITC experiments, the Core Tissue Culture team for cell provision, Pratik Saxena for hESC-line generation support, and Andrew Leinster, Kelli Ryan, the Oncology in vivo team, John Peverill and the Animal Sciences and Technologies team for support with the xenograft mouse study.

## Author contributions

S.E.C. designed & performed AAV experiments and wrote the paper. S.L. designed & performed phage display experiments. L.V. designed & performed HTRF assays. R.B.D. & L.B. carried out protein expression. L.B. designed & performed split transcription factor luciferase assays. D.G.R. and A.S. designed & performed kinetic experiments. S.J.S. designed & performed endogenous gene expression experiments. C.S., C.G. & J.S. designed & performed in vitro kill switch experiments. T.I.M.M. designed & performed in vivo kill switch experiments. N.J.T. conceived of and designed the research. All authors edited the paper.

## Competing interests

All authors are current or former employees of AstraZeneca and may hold stocks or shares. Authors L.B., R.B.D., S.L., D.G.R., A.S., L.V., C.S., and N.J.T. are inventors on WIPO patent application WO2021009692A1. The authors declare no other competing interests.
