## [Peer Review File · Nature Communications]

Reviewers' Comments:

Reviewer #1:

Remarks to the Author:

Tigue and colleagues developed a novel molecular glue based on a clinically approved small molecule Simeprevir, a newly developed antibody-like molecule, and an engineered protease domain from Hepatitis C virus (NS3/4A). The authors demonstrated utility in a number of applications of interest in mammalian synthetic biology, including driving IL-2 expression endogenously, driving exogenous genes, developing a mammalian kill switch and demonstrating utility of small-molecule regulated apoptosis in a tumor bearing mouse model. The advantages of this system include the use of an already clinically approved small molecule, the use of exogenous proteins for the CID switch, and the very clean mammalian kill switch data. As a body of work this is an impressive contribution and will add to the growing literature on engineered CID modules. There are two requested revisions detailed below:

Revisions:

1. Reconsider the kinetic characterization of the engineered molecular glue in terms of existing models for ternary complex formation – the listed K_d in Figure 1 is an apparent K_d valid only under the exact concentrations of the FNIII and protease used. The explanation for the dumbbell like binding equilibria observed in the HTRF binding screens shown in Figure 1 & Supp Fig 3 may derive from the ternary complex equilibria – at high concentrations of both NS3/4A and FNIII each can bind Simeprevir independently, leading to lower ternary complex formation. Molecular glues cannot usually be fit to saturable binding isotherms like that presented in Figure 1. The two references that can help define the binding isotherms and design of additional experiments, if needed to describe the binding isotherms, are:

o Cao et al "Defining molecular glues with a dual-nanobody cannabidiol sensor" Nature Communications 2022

o Douglass et al, "A comprehensive mathematical model for three-body binding equilibria" JACS 2013

2. Reframe the introduction for the novelty design/engineering of chemically induced dimerization modules; there is literature that has been missed (and reviewed thoroughly in Leonard and Whitehead Current Opinion in Biotechnology 2022) that should be included in the introduction.

o Kang et al. ("COMBINES-CID:..." JACS 2019) developed a previous strategy called COMBINES-CID for sandwiching a known ligand (in their case they didn't start with a known ligand binding domain) using nanobodies.

o More similarly to the present contribution, Guo et al ("Design of a methotrexate-controlled chemical dimerization system..." Nature Communications 2021) developed a binder against a known methotrexate binding domain, although the application area was distinct.

o In the introduction, the authors list a mandipropamid sensor as a "natural CID" line 30 – it is not; it is an engineered sensor derived from a natural plant CID. Additionally, this engineering platform was recently extended to build CID modules for human drugs (in this case, the illicit variety) – see Beltran et al Nature Biotechnology 2022.

I think these additional references would help the narrative but not shrink the current contribution – the data presented in figures 2-4 are among the strongest I've seen for mammalian synthetic biology.

Tim Whitehead (I'm including my name here as I've asked the authors to cite some of my work in the introduction, which I don't typically do).

Reviewer #2:

Remarks to the Author:

The manuscript describes the construction of chemically induced protein dimerization system and its use in cell biological and bioengineering applications.

The main differentiating feature of this paper from previous studies is the very judicious choice of the anchor domain and the chemical compound from the point of view of biological orthogonality. Additional modifications of the anchor domain (viral protease) are done to reduce its potential

unintended influence on biological systems. Authors use the anchor:small molecule complex as the bait in the phage display selection system and isolate lid domains that complete the novel CID system. The CID system is tested in a range of cellular and organismal models and shown to perform well and in fact superior to the most used rapamycin-controlled protein dimerization system.

This is a very well planned, executed and written study and is an example of how synthetic biology can deliver enabling biological tools. I recommend publication of the paper after a couple of issues are addressed.

1) The authors analyse the developed CID system biophysically and establish that in the presence of a ligand the K_d of the system is in single digit nM. However they provide no estimate of the affinity of the system in the ligand free state which is quite important for understanding its performance parameters.

2) Authors describe construction of transcription inducing system (line 141) where they create repeats of the anchor and lid domains to achieve larger response. While the data shown in figure 2C is very compelling it is also very puzzling as one would expect that cooperativity would drive the increase in the avidity between fusion lid and fusion anchor domains. That brings us back to the question raised above as the system could only work if affinity between components in the absence of the ligand is very low. Therefore I would strongly encourage the authors to perform ICT (or another methods that can quantify low affinity interactions) analysis of the CID in the absence of the ligand. Even if the interaction is not detectable it would give an estimate of the low affinity limit. One also could do the same for the multimerised units where one would expect to see some affinity. It would also be interesting and important to understand the apparent K_d gains in such polymeric systems in the presence of the ligand.

Minor points:

3) The fonts in figures are very small and hard to read

4) Line 159 – concentrations should be molar to enable the reader to relate this information to the earlier determined system's K_d s.

5) The authors should provide the sequences of the constructs they used as it would help to understand their designs. While one can find the patents in now published patent covering this work including it into the paper would help the reader.

Reviewer #3:

Remarks to the Author:

Chin et al report a new chemical dimerization system that has the potential to be easily adopted in gene and cell therapy. The small molecule is a clinically approved antiviral compound called simeprevir. The first protein in this heterodimerization system is the simeprevir target, the NS3/4A protease, from hepatitis C virus, and the second protein is identified by phage display selections for HCV NS3/4A PR:Simeprevir complex-specific binding. The authors demonstrated the ability of the dimerization system to control gene expression and apoptosis in cultured cells and tumor shrinkage via apoptosis in a xenograft mouse model. I find that the designing of dimerization system based on clinically approved small molecules is an interesting and important strategy for realizing the therapeutic potential of chemical dimerizers. I also find most conclusions are well-supported. Some minor concerns are:

1. In the peptide cleavage assay in FigureS1 b, the activity seems to increase after at 10^{-6} M. There is no information about the assay in experiment or figure caption for interpretation of this figure. What peptide was used? How is 10^{-6} M in comparison to the cellular concentration? In other words, why stop at 10^{-6} M? These questions are important because using a protease in a dimerization system is concerning.

2. Figure 3 why assessing cell death at 100 nM while Caspase 3 activity at 10nM?

3. Figure 4 why only observe the untreated 40 days while the two 60 days? What is reason for

choosing 200 mg/kg as the dosage? Is the estimated concentration close to mole concentration used in Figure 3? What does each line in the figure represent?
4. Fonts in figures are way too small, very hard to read.

Reviewer #4:

Remarks to the Author:

In this manuscript, Chin et al develop a CID molecular switch based on simeprevir and the HPV NS3/4A protease. Overall, their experimental design and the data showing the usefulness of the system as a research tool for tuning gene expression are sound. However, the suitability of the system for control of cell and gene therapies remains questionable.

To demonstrate that the proposed system is suitable for control of cell and gene therapies, the system should be (at least in vitro) tested in the relevant cell models (e.g., CAR-T cells) instead of the very easy to handle cancer cell lines that are currently used in the study.

The in vitro kill switch experiments are interesting but lack experimental details.

- Figure 3b: For how long were the cells treated when the images were taken? I assume that the treatment time was the same across the different cell lines? How many cells were plated? This all needs further clarification.

- Figure 3b: the images are not convincing. Immunofluorescence staining for cleaved caspase-3 or another apoptosis marker should be performed.

- Figure 3c: the fold induction should be calculated relative to untreated parental HEK293T cells so that the potential off-target effects of simeprevir can be determined. From the current graph, it seems as if there is an induction of caspase 3 activity in the absence of the kill switch too. The authors should comment on this.

- Similar to the above, Figure 3d lacks the "no simeprevir" condition. This should be added to exclude that the compound itself already induces apoptosis. This is crucial if the system is going to be used in a therapeutic setting.

In its current state, the PDX experiment convincingly shows that simeprevir can also be used to efficiently switch HT29 cells in vivo. I understand that, as proof-of-principle, the authors performed the experiment with only one cell line. However, to show the potential of the switch for therapeutic purposes requires additional analyses. For example, experimental groups where HT29 cells without the kill switch are transplanted and treated with simeprevir or not should be included to demonstrate that the observed effects are exclusively on target. Furthermore, the authors should demonstrate that there are no signs of toxicity in mice treated with simeprevir (body weight measurements, histological analyses of organs, etc).

Reviewer #1

1. Reconsider the kinetic characterization of the engineered molecular glue in terms of existing models for ternary complex formation – the listed K_d in Figure 1 is an apparent K_d valid only under the exact concentrations of the FNIII and protease used. The explanation for the dumbbell like binding equilibria observed in the HTRF binding screens shown in Figure 1 & Supp Fig 3 may derive from the ternary complex equilibria – at high concentrations of both NS3/4A and FNIII each can bind Simeprevir independently, leading to lower ternary complex formation. Molecular glues cannot usually be fit to saturable binding isotherms like that presented in Figure 1. The two references that can help define the binding isotherms and design of additional experiments, if needed to describe the binding isotherms, are:

- o Cao et al “Defining molecular glues with a dual-nanobody cannabidiol sensor” Nature Communications 2022
- o Douglass et al, “A comprehensive mathematical model for three-body binding equilibria” JACS 2013

Response:

Thank you for pointing out that the dumbbell-like equilibria observed in HTRF might be due to simeprevir binding to the PRSIM Tn3 at high Tn3 concentrations in addition to/rather than detection reagent depletion. We carried out an additional HTRF experiment to test this hypothesis, which has been added to the manuscript as Supplementary Figure 4. In this experiment, we fixed the concentrations of HCV PR and PRISM_23 at 5 nM and 6 nM, respectively, and measured complex formation in the presence of a titration of simeprevir from 57 fM to 7.4 μ M. As expected, complex formation increases as a function of simeprevir concentration until reaching a plateau corresponding to maximum complex formation limited by the concentrations of the protein components. At concentrations of simeprevir >5 nM, HCV PR will be saturated, given the estimated K_D for simeprevir binding to HCV PR of 1 pM determined by ITC. As simeprevir concentration increases further, and is in excess compared to the HCV PR concentration, excess simeprevir will be available for binding PRSIM_23. If simeprevir were binding to PRSIM_23 at high concentrations, we would expect to see a decrease in signal due to competition for binding PRSIM_23 between excess simeprevir in solution and the HCV PR-simeprevir complex, but we do not see any such inhibition. Thus, the most likely explanation for the dumbbell-like equilibria in Figure 1 is reagent depletion. We have edited the legend for Figure 1 (line 891) to highlight this.

This experiment also further suggests that any interaction between simeprevir and PRSIM_23 alone is very weak (> 7.4 μ M; compare this to the affinity of simeprevir for HCV PR which we determined by ITC to be ~1 pM), suggesting that our molecular switch is acting more like the rapamycin-based CID than via a molecular glue-type mechanism where the small molecule interacts with both protein components.

Thank you for highlighting relevant papers to help guide design of additional experiments to more fully characterize the kinetics and affinities defining formation of the ternary complex which comprises our switch. Given that our system is behaving similarly to the rapamycin-based system, we have used Cao et al Figure 7 and Table 1 as a guide for designing additional experiments, as these authors have characterized several different CIDs with rapamycin-like binding interactions. In particular, the authors investigate the following interactions experimentally, determining affinities where possible:

- 1) Protein 1 – small molecule
- 2) Protein 2 – small molecule
- 3) Protein 1 – protein 2 in the absence of small molecule
- 4) Protein 2 binding to preformed protein 1-small molecule complex

For our system, these correspond to:

- 1) HCV PR + simeprevir
- 2) PRSIM_23 + simeprevir
- 3) HCV PR + PRSIM_23
- 4) HCV PR-simeprevir + PRSIM_23

The original manuscript addressed interactions 1&3; in the revised manuscript, we have added experiments to address interactions 2 & 4, and therefore, have now investigated each of these interactions experimentally, as described below:

- 1) The affinity of the HCV PR - simeprevir interaction was determined by ITC, and is shown in Supplementary Figure 1.
- 2) This interaction has been investigated in the HTRF experiment described above, added to the manuscript as Supplementary Figure 4. This is similar to the approach used in Foight, G.W. et al. Multi-input chemical control of protein dimerization for programming graded cellular responses. *Nat Biotechnol* **37**, 1209-1216 (2019) to investigate the equivalent interaction in their CID system.
- 3) We have investigated whether PRISM_23 and HCV PR interact in the absence of simeprevir by SPR (similar to experiments in Hill, Z.B., Martinko, A.J., Nguyen, D.P. & Wells, J.A. Human antibody-based chemically induced dimerizers for cell therapeutic applications. *Nat Chem Biol* **14**, 112-117 (2018)). We observe weak binding between PRSIM_23 and HCV PR. This can be fit to a 1:1 binding model and gives an estimated K_D of 6.2 μ M. This has been added to the manuscript as Figure 1e.
- 4) The affinity of PRSIM_23 for the preformed HCV PR-simeprevir complex was determined to be 6.1 nM by SPR as shown in Figure 1d. As Reviewer #1 highlights, this K_D is an apparent K_D only valid under these experimental conditions. Accordingly, we have changed K_D to apparent K_D in the text.

Based on these results, we are now able to calculate the cooperativity of our system, which is a key parameter describing CID systems and is determined by the ratio between 3) and 4) above. We estimate the cooperativity of our system to be \sim 1000, which is comparable to other CID modules in the literature. We have edited the text (line 131) to reflect this.

2. Reframe the introduction for the novelty design/engineering of chemically induced dimerization modules; there is literature that has been missed (and reviewed thoroughly in Leonard and Whitehead Current Opinion in Biotechnology 2022) that should be included in the introduction.

o Kang et al. ("COMBINES-CID:..." JACS 2019) developed a previous strategy called COMBINES-CID for sandwiching a known ligand (in their case they didn't start with a known ligand binding domain) using nanobodies.

o More similarly to the present contribution, Guo et al ("Design of a methotrexate-controlled chemical dimerization system..." Nature Communications 2021) developed a binder against a known methotrexate binding domain, although the application area was distinct.

o In the introduction, the authors list a mandipropamid sensor as a "natural CID" line 30 – it is not; it is an engineered sensor derived from a natural plant CID. Additionally, this engineering platform was recently extended to build CID modules for human drugs (in this case, the illicit variety) – see Beltran et al Nature Biotechnology 2022.

I think these additional references would help the narrative but not shrink the current contribution – the data presented in figures 2-4 are among the strongest I've seen for mammalian synthetic biology.

Response:

Thank you for highlighting literature that we had missed. We have updated the introduction and incorporated the suggested references (Ref. 6, 8, 11 and 13 in updated reference list) .

Reviewer #2

1) The authors analyse the developed CID system biophysically and establish that in the presence of a ligand the K_D of the system is in single digit nM. However they provide no estimate of the affinity of the system in the ligand free state which is quite important for understanding its performance parameters.

Response:

Figure 1c and Supplementary Figure 3 show that the PRSIM molecules do not bind to the HCV NS3/4A PR alone (in the absence of simeprevir) in an HTRF assay at concentrations up to 500 nM. We acknowledge, however, that this does not rule out a weaker affinity interaction. Therefore, we have investigated this interaction at higher concentrations using Biacore, and added this data to the manuscript as Figure 1e. We observe weak binding in this assay, with a K_D value of 6.2 μ M. We can use this value, together with the affinity calculated in Figure 1d to estimate the cooperativity of the CID to be 1000, which compares favourably to similar CID systems in the literature (e.g., examples in Cao et al “Defining molecular glues with a dual-nanobody cannabidiol sensor” Nature Communications 2022). We have edited the text (line 131) to include a discussion of the cooperativity of our CID system.

2) Authors describe construction of transcription inducing system (line 141) where they create repeats of the anchor and lid domains to achieve larger response. While the data shown in figure 2C is very compelling it is also very puzzling as one would expect that cooperativity would drive the increase in the avidity between fusion lid and fusion anchor domains. That brings us back to the question raised above as the system could only work if affinity between components in the absence of the ligand is very low. Therefore I would strongly encourage the authors to perform ICT (or another methods that can quantify low affinity interactions) analysis of the CID in the absence of the ligand. Even if the interaction is not detectable it would give an estimate of the low affinity limit. One also could do the same for the multimerised units where one would expect to see some affinity. It would also be interesting and important to understand the apparent K_D gains in such polymeric systems in the presence of the ligand.

Response:

In Figure 2c, only the PRSIM molecules are multimeric. The HCV PR-AD modules are monomeric, and thus would be expected to each bind independently to a single PRSIM molecule - this is consistent with the additive, rather than cooperative effect, that we observe in this assay. Investigation of the binding kinetics of complexes formed between multimeric HCV-PR and multimeric PRSIM molecules would be interesting but is beyond the scope of this manuscript.

3) The fonts in figures are very small and hard to read

Response:

Thank you for pointing this out. We have increased the fonts in figures for improved readability.

4) Line 159 – concentrations should be molar to enable the reader to relate this information to the earlier determined system’s K_D s.

Response:

The concentrations quoted in pg/ml (as is the convention for cytokines) are for the maximal levels of IL-2 induced by simeprevir via the PRSIM_23-based split transcription factor; this is unrelated to the apparent K_D for the ternary HCV PR-simeprevir-PRSIM_23 complex quoted earlier in the text. For clarity, we have added the EC50 for simeprevir from this experiment to the text (line 184), which is related to the ternary complex affinity.

5) The authors should provide the sequences of the constructs they used as it would help to understand their designs. While one can find the patents in now published patent covering this work including it into the paper would help the reader.

Response:

Thank you. As requested we have added construct sequences as Supplementary Table 4.

Reviewer #3

1. In the peptide cleavage assay in Figure S1 b, the activity seems to increase after at 10^{-6} M. There is no information about the assay in experiment or figure caption for interpretation of this figure. What peptide was used? How is 10^{-6} M in comparison to the cellular concentration? In other words, why stop at 10^{-6} M? These questions are important because using a protease in a dimerization system is concerning.

Response:

We have added details of the peptide cleavage assay to the figure legend for Supplementary Figure 1. As the reviewer notes, there is residual activity of the S139A mutant at very high ($\geq \mu\text{M}$) concentrations of HCV PR. The S139A mutant has been shown to eliminate cleavage of MAVS (Ref. 17), a known host protein cleavage target of HCV PR, and a synthetic sensor substrate (Ref. 19), suggesting that when expressed in a cell, the levels of this mutant are not high enough for the residual activity of the S139A variant to cause detectable levels of substrate cleavage. However, we have changed "will" to "should" in line 287 of the discussion to reflect that we cannot completely rule out residual cleavage activity of the S139A mutant in the cell.

2. Figure 3 why assessing cell death at 100 nM while Caspase 3 activity at 10nM?

Response:

Thank you for the question. We also observe rapid cell killing at 10 nM, so we have replaced the images taken after addition of 100 nM simeprevir with images taken after addition of 10 nM simeprevir for consistency.

3. Figure 4 why only observe the untreated 40 days while the two 60 days? What is reason for choosing 200 mg/kg as the dosage? Is the estimated concentration close to mole concentration used in Figure 3? What does each line in the figure represent?

Response:

The untreated mice had to be culled due to tumour volume limits or welfare reasons such as ulceration. There were no signs of toxicity in the treated group (body weight measurements have been added as Supplementary Figure 12 to support this statement), so we were able to monitor this group for a longer time period to determine whether the tumours regrew. The 200 mg/kg dose was chosen based on PK studies with simeprevir which showed that at this dosage level the exposure of simeprevir was higher than the EC50 value measured in vitro for induction of the kill switch. This simeprevir PK data has been added to the supplementary information (Supplementary Figure 11). Each line represents an individual mouse; the figure legend has been edited to include this information.

4. Fonts in figures are way too small, very hard to read.

Response:

Thank you for highlighting this. We have increased the fonts in figures for improved readability.

Reviewer #4

To demonstrate that the proposed system is suitable for control of cell and gene therapies, the system should be (at least *in vitro*) tested in the relevant cell models (e.g., CAR-T cells) instead of the very easy to handle cancer cell lines that are currently used in the study.

Response:

We have added two additional figures to demonstrate activity of the kill switch in therapeutically relevant cells. In Figure 3h, we show that the kill switch is active in the human embryonic stem (ES) cell line Sa121; hES cells are of significant interest as therapeutic modalities in the field of regenerative medicine. In Figure 3i, we demonstrate that the kill switch is active in primary human CD8+ T cells. T-cell based therapies, including tumor infiltrating lymphocytes, TCR-T cells and CAR-T cells, are being widely explored as therapies to treat cancer.

The *in vitro* kill switch experiments are interesting but lack experimental details.

- Figure 3b: For how long were the cells treated when the images were taken? I assume that the treatment time was the same across the different cell lines? How many cells were plated? This all needs further clarification.
- Figure 3b: the images are not convincing. Immunofluorescence staining for cleaved caspase-3 or another apoptosis marker should be performed.
- Figure 3c: the fold induction should be calculated relative to untreated parental HEK293T cells so that the potential off-target effects of simeprevir can be determined. From the current graph, it seems as if there is an induction of caspase 3 activity in the absence of the kill switch too. The authors should comment on this.
- Similar to the above, Figure 3d lacks the “no simeprevir” condition. This should be added to exclude that the compound itself already induces apoptosis. This is crucial if the system is going to be used in a therapeutic setting.

Response:

Figure 3b - The figure legends and methods have been updated with details of treatment time and number of cells plated.

As an alternative to the suggested immunofluorescence experiment, we have confirmed induction of apoptosis by demonstrating simeprevir-induced cleavage of the apoptosis effectors Caspase 9 and Caspase 3 in these cells by Western blot (Supplementary Figure 9).

Figure 3c & 3d – As suggested, the fold induction has now been calculated relative to untreated parental cells. There is no statistically significant difference between induction of Caspase 3 activity in untreated parental cells vs parental cells treated with 10 nM simeprevir. Furthermore, simeprevir is a clinically approved compound with an acceptable safety profile for daily dosing, suggesting that it is well-suited for use in a therapeutic setting.

In its current state, the PDX experiment convincingly shows that simeprevir can also be used to efficiently switch HT29 cells *in vivo*. I understand that, as proof-of-principle, the authors performed the experiment with only one cell line. However, to show the potential of the switch for therapeutic purposes requires additional analyses. For example, experimental groups where HT29 cells without the kill switch are transplanted and treated with simeprevir or not should be included to demonstrate that the observed effects are exclusively on target. Furthermore, the authors should demonstrate that there are no signs of toxicity in mice treated with simeprevir (body weight measurements, histological analyses of organs, etc).

Response:

The *in vitro* data presented in Figures 3d and e indicate that there is no off-target killing of parental HT29 cells that do not contain the kill switch. As such, it was not considered ethical to include groups with parental HT29 cells in the *in vivo* study. We see no effect on body weight of simeprevir

treatment and have added the body weight measurements from the in vivo study to the Supplementary Information as Supplementary Figure 12.

REVIEWERS' COMMENTS

Reviewer #1 (Remarks to the Author):

The authors have satisfied all technical concerns I originally had with the manuscript; well done!

The engineered system appears to function like a molecular ratchet rather than a molecular glue; one additional recent paper from my group solved the algebraic closed form model for this class of CID mechanisms, which should help in future efforts for the forward engineering of PRSIM in engineered cells:

Steiner PJ et al (2022) "A closed form model for molecular ratchet-type chemically induced dimerization modules" Biochemistry

Reviewer #2 (Remarks to the Author):

The authors addressed the minor concerns that I had and provided additional experimental data that elucidates the operational mechanism of the developed CID. The manuscript is now acceptable for publication. I congratulate the authors on a very nice study !

Reviewer #3 (Remarks to the Author):

The authors have addressed my concerns.

Reviewer #4 (Remarks to the Author):

My comments have been satisfactorily addressed.